# Multi-ancestry genome-wide association analyses improve resolution of genes and pathways influencing lung function and chronic obstructive pulmonary disease risk

**A list of authors and their affiliations appears at the end of the paper**

Lung-function impairment underlies chronic obstructive pulmonary disease (COPD) and predicts mortality. In the largest multi-ancestry genome-wide association meta-analysis of lung function to date, comprising 588,452 participants, we identified 1,020 independent association signals implicating 559 genes supported by ≥2 criteria from a systematic variant-to-gene mapping framework. These genes were enriched in 29 pathways. Individual variants showed heterogeneity across ancestries, age and smoking groups, and collectively as a genetic risk score showed strong association with COPD across ancestry groups. We undertook phenome-wide association studies for selected associated variants as well as trait and pathway-specific genetic risk scores to infer possible consequences of intervening in pathways underlying lung function. We highlight new putative causal variants, genes, proteins and pathways, including those targeted by existing drugs. These findings bring us closer to understanding the mechanisms underlying lung function and COPD, and should inform functional genomics experiments and potentially future COPD therapies.

Lung-function abnormality predicts mortality and is a diagnostic criterion for chronic obstructive pulmonary disease (COPD)[1], which is the most prevalent respiratory disease globally[2] and lacks disease-modifying treatments. Although smoking and other environmental risk factors for COPD are well known and genetic susceptibility is recognized, the molecular pathways underlying COPD are incompletely understood. As with other complex traits there has been a lack of ancestral diversity in genome-wide association studies (GWAS)[3] of lung function[4–6]. Multi-ancestry studies improve the power and fine-mapping resolution of GWAS and increase the prospects for prediction, prevention, diagnosis and treatment in diverse populations[3,4,7].

Understanding of the genes, proteins and pathways involved in disease-related traits underpins modern drug development. A high yield of genetic-association signals, improved signal resolution and integration with functional evidence assist confident identification of causal genes as well as the variants and pathways that impact gene function and regulation. Although datasets and in silico tools to connect GWAS signals to causal genes are improving, the findings from different datasets and tools have lacked consensus[8,9], highlighting a need for frameworks to integrate functional evidence types and compare findings[10].

Aggregation of lung-function-associated genetic variants into a genetic risk score (GRS) provides a tool for COPD prediction[5]. When a GRS comprises many variants, partitioning the GRS according to the biological pathways the variants influence could provide a tool to explore their aggregated consequences across different traits through phenome-wide association studies (PheWAS). Just as PheWAS of individual genetic variants predicts the consequences of perturbations of specific protein targets, informing assessment of drug efficacy, drug safety and drug repurposing[11], PheWAS of pathway-partitioned GRS

✉e-mail: nick.shrine@leicester.ac.uk; martin.tobin@leicester.ac.uk

**Fig. 1 | Study overview. a**, Discovery meta-analysis. *For signals present in more than one trait, the signal is only counted once (for the most significant trait). **b**, Pathway analyses, GRS analyses and PheWAS studies.

could inform the understanding of the consequences of perturbations of specific pathways.

Through the largest global assembly of lung-function genomics studies to date we: (1) undertook a multi-ancestry GWAS meta-analysis of lung-function traits in 588,452 individuals to detect novel signals, improve fine mapping and estimate heterogeneity in allelic effects attributable to ancestry; (2) tested whether lung-function signals are age- or smoking-dependent, and assessed their relationship to height; (3) investigated cell-type and functional specificity of lung-function association signals; (4) fine-mapped signals through annotation-informed credible sets, integrating functional data such as respiratory cell-specific chromatin accessibility signatures; (5) applied a consensus-based framework to systematically investigate and identify putative causal genes, integrating eight locus-based or similarity-based criteria; (6) developed and applied a GRS for the ratio of forced expiratory volume in 1 s ($FEV_1$) to forced vital capacity (FVC) in different ancestries in the UK Biobank and COPD case–control studies; and (7) applied PheWAS to individual variants, GRS for each lung-function trait and GRS partitioned by pathway. Through these approaches, we aimed to detect novel lung-function signals and putative causal genes as well as provide new insights into the mechanistic pathways underlying lung function, some of which may be amenable to drug therapy.

## Results

We undertook genome-wide association analyses of $FEV_1$, FVC, $FEV_1$/FVC and peak expiratory flow rate (PEF) from 49 cohorts (Methods and Supplementary Tables 1,2). Our study of up to 588,452 participants comprised individuals of African (AFR; $n = 8,590$), American/Hispanic (AMR; $n = 14,668$), East Asian (EAS; $n = 85,279$), South Asian (SAS; $n = 4,270$) and European ancestry (EUR; $n = 475,645$; Supplementary Fig. 1a,b). In cohort-specific analyses we adjusted for age, age squared, sex and height, accounting for population structure and relatedness (Methods

and Supplementary Tables 2–4), and then applied genomic control using the linkage disequilibrium (LD) score regression intercept[12]. After filtering and meta-analysis across multi-ancestry cohorts, 66.8 million variants were available in each of four lung-function traits, with genomic inflation factors $\lambda$ of 1.025, 1.022, 0.984 and 0.996 for $FEV_1$, FVC, $FEV_1$/FVC and PEF, respectively (Supplementary Figs. 2,3 and Supplementary Table 5).

### 1,020 signals for lung function

After excluding eight signals associated with smoking behavior (Supplementary Table 26) and combining signals that co-localized across traits, we identified 1,020 distinct signals for lung function using a stringent threshold of $P < 5 \times 10^{-9}$ (ref.[13]; Fig. 1a). Of these, 713 are novel with respect to the signals and studies described in the Supplementary Note (Supplementary Table 6). These 1,020 signals explain 33.0% of $FEV_1$/FVC heritability (21.3% for $FEV_1$, 17.3% for FVC and 21.4% for PEF; Methods).

To facilitate fine mapping, we included larger, more diverse populations than previous lung-function GWAS. We performed multi-ancestry meta-regression with MR-MEGA[7], which incorporates axes of genetic ancestry as covariates to model heterogeneity (Methods). We then incorporated functional annotation for chromatin accessibility and transcription-factor binding sites in respiratory-relevant cells and tissues, and enriched genomic annotations[14] to weight prior causal probabilities of association for putative causal variants (Methods). Overall reductions in credible set size and higher maximum posterior probabilities of association for the most likely causal variants were evident after multi-ancestry meta-regression and after functional annotations were incorporated (Supplementary Fig. 4). Following fine mapping, 438 (43%) signals had a single putative causal variant (posterior probability > 50%) and the median credible set size was nine variants (Supplementary Note).

Of the 960 sentinels represented in ≥7 cohorts, 109 signals showed heterogeneity attributable to ancestry ($P_{Het} < 0.05$; Supplementary

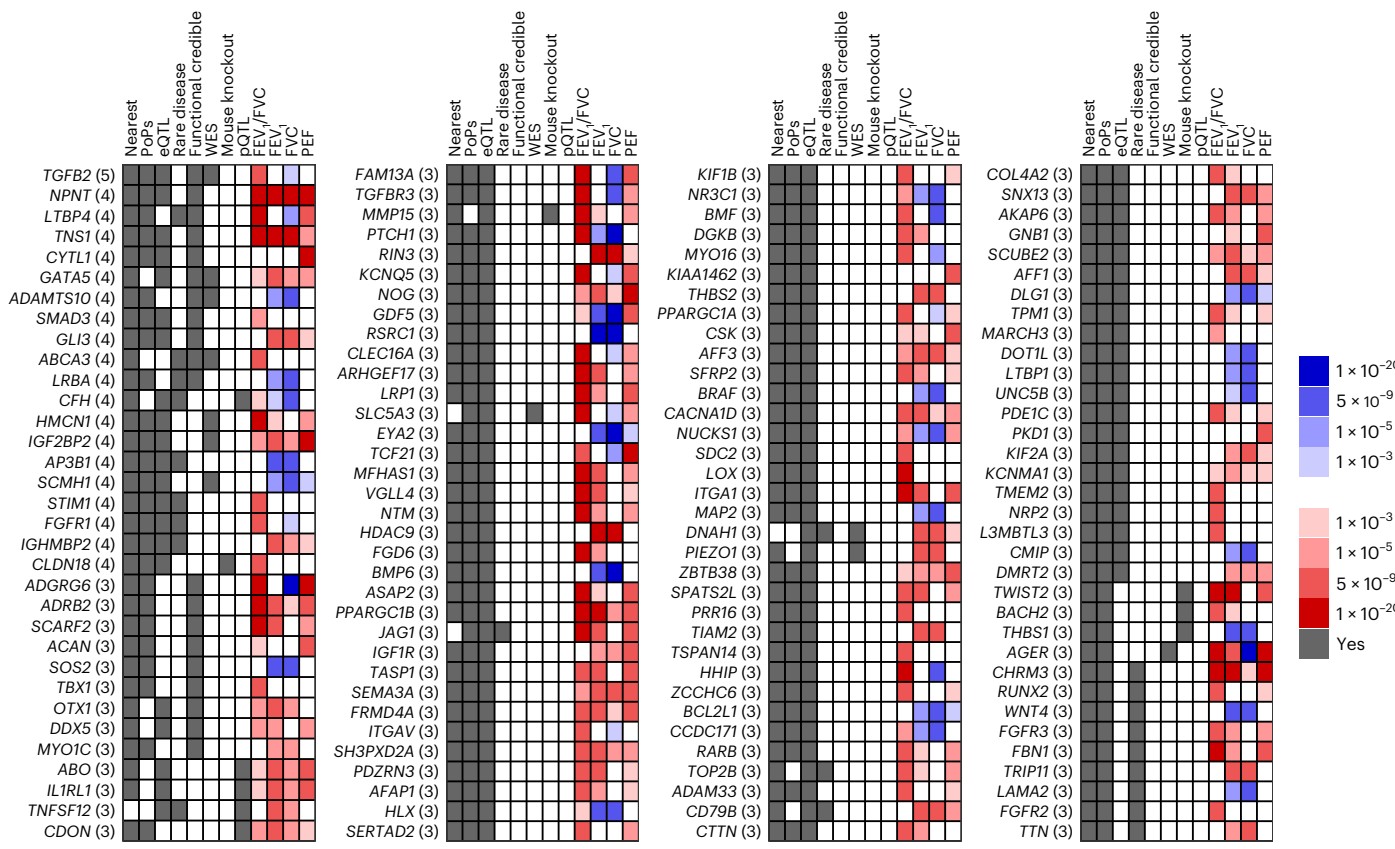

**Fig. 2 | 135 genes prioritized with ≥3 variant-to-gene criteria.** The number of variant-to-gene criteria implicating the gene is in brackets after the gene name. The gray in the first eight columns indicates that at least one variant implicates the gene as causal via the evidence for that column. The last four columns indicate the level of association of the most significant variant implicating the gene as causal with respect to the $FEV_1/FVC$ decreasing allele; red indicates that this association is in the same direction of effect as the $FEV_1/FVC$ decreasing allele and blue indicates the opposite direction with the shade indicating $P$ < the corresponding value in the legend.

Fig. 5 and Supplementary Table 7), which was more than expected (binominal test, $P = 3.93 \times 10^{-15}$). Among these, five signals (rs9393688, rs28574670 (*LTBP4*), rs7183859 (*THSD4*), rs59985551 (*EFEMP1*) and rs78101726 (*MECOM*)) showed significant ancestry-correlated heterogeneity (Bonferroni correction for 960 signals tested, $P_{Het} < 5.21 \times 10^{-5}$; Supplementary Fig. 6a–e). The intronic variant rs7183859 in *THSD4*, which we previously implicated in lung function[15], showed larger effect-size estimates in non-EUR ancestries and in particular AFR ancestries ($P_{Het} = 3.33 \times 10^{-5}$; Supplementary Fig. 6c).

We examined associations of lung-function-associated SNPs in children's cohorts (Supplementary Table 8) and tested for differences in the estimated effect sizes of lung-function-associated SNPs between children and adults as well as between ever-smokers and never-smokers in EUR individuals (Methods). The effect-size estimates between children and adults were correlated ($r$ from 0.51 for $FEV_1/FVC$ to 0.79 for $FEV_1$; Supplementary Fig. 7), although 113 signals showed nominal evidence ($P < 0.05$) of age-dependent effects (more than expected, binomial $P = 2.56 \times 10^{-13}$). Three signals (rs7977418 (*CCDC91*), rs34712979 (*NPNT*) and rs931794 (*HYKK*)) showed age-dependent effects (Bonferroni-corrected $P < 4.64 \times 10^{-5}$; Supplementary Table 9). We observed nominal evidence ($P < 0.05$) of smoking-dependent effects for 69 of 1,020 signals (Supplementary Fig. 8), more than expected (binomial $P = 0.0079$). The intronic SNP rs7733410 in *HTR4*, a signal we previously reported for lung function[15], showed a 76.2% larger effect on $FEV_1$ in ever-smokers compared with never-smokers ($P = 4.09 \times 10^{-5}$; Supplementary Table 10). As height is a determinant of lung growth, we compared height and lung-function associations, and tested the impact of additional height adjustments for sentinel SNPs. We found

no correlation between estimated effect sizes for height and lung function (Supplementary Fig. 9), and the addition of height squared and height cubed covariates had little impact on effect-size estimates (Supplementary Fig. 10).

**Cell-type and functional specificity**

We assessed whether our association signals were enriched for regulatory or functional features in specific cell types. Using stratified LD-score regression[16], we found enrichment of all histone marks tested (H3K27ac, H3K9ac, H3K4me3 and H3K4me1) in lung- and smooth-muscle-containing cell lines (Supplementary Table 11). Using GARFIELD[17] we assessed for enrichment of our signals for DNase I hypersensitivity sites and chromatin accessibility peaks, showing enrichment in a wide variety of cell types, including higher enrichment in both fetal and adult lung and blood for $FEV_1$, $FEV_1/FVC$ and PEF as well as fibroblast enrichment for FVC (Supplementary Fig. 11a). Our signals were enriched for transcription-factor footprints in fetal lung for $FEV_1$, $FEV_1/FVC$ and PEF, for footprints in skin for FVC and also in blood for PEF (Supplementary Fig. 11b). Genic annotation enrichment patterns were similar across all traits, with enrichment mainly in exonic, 3′ UTR and 5′ UTR regions (Supplementary Fig. 11c). For all traits, we saw enrichment for transcription start sites, weak enhancers, enhancers and promoter flanks, with cell types for weak enhancer enrichment including endothelial cells for $FEV_1$, $FEV_1/FVC$ and PEF (Supplementary Fig. 11d). For transcription-factor binding sites, we observed a similar enrichment pattern across all of the lung-function traits, with the largest fold enrichment observed for endothelial cells (Supplementary Fig. 11e). Our signals were enriched for assay for transposase-accessible chromatin

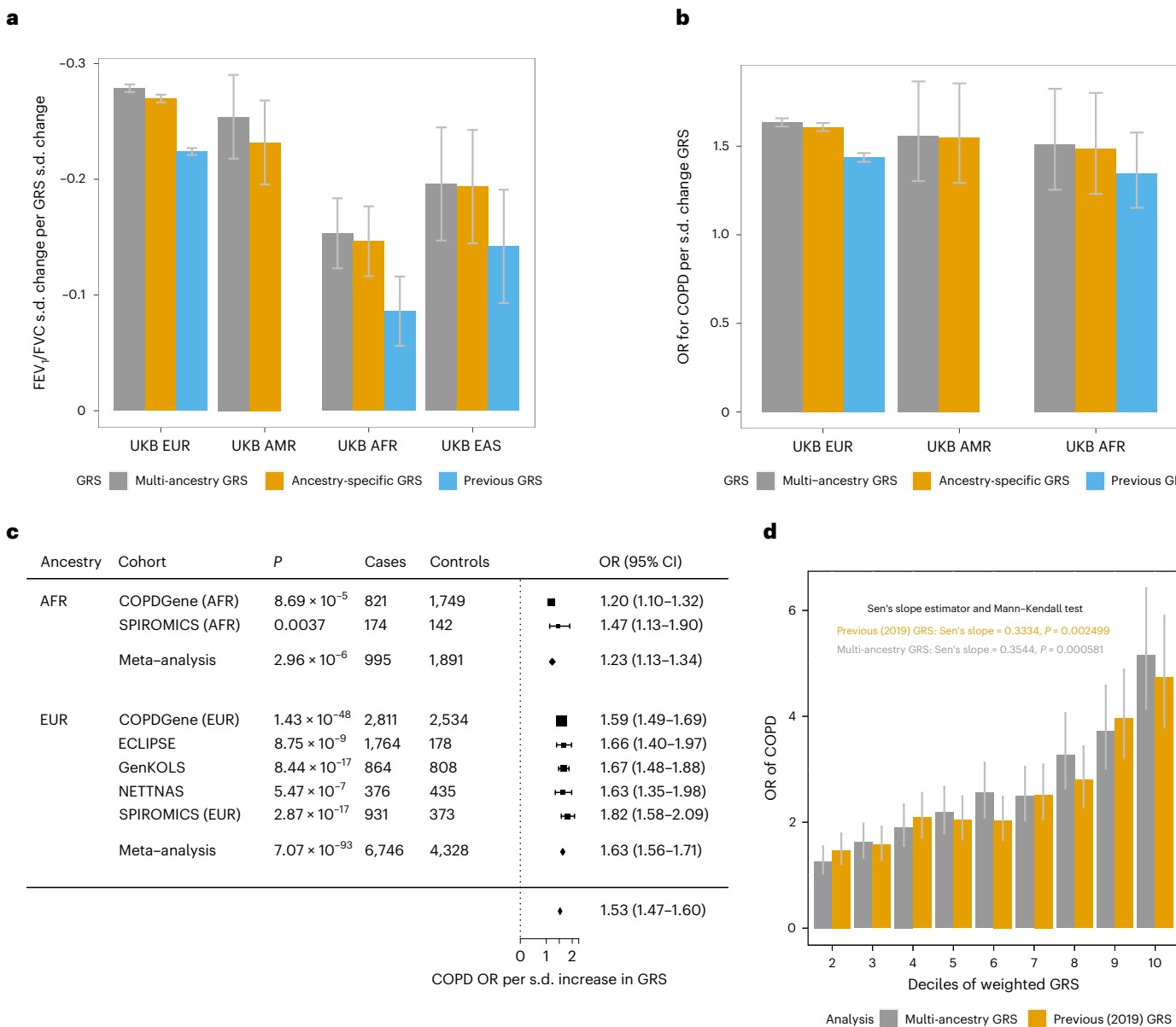

**Fig. 3 | GRS performance. a**, Prediction performance of three GRSs across ancestry groups for $FEV_1/FVC$ shown as the s.d. change in $FEV_1/FVC$ per s.d. increase in GRS for individuals in the UK Biobank grouped according to ancestry. Sample sizes: AFR, $n = 4,227$; AMR, $n = 2,798$; EAS, $n = 1,564$; and EUR, $n = 320,656$. **b**, Prediction performance of three GRSs for COPD shown as COPD odds ratio per s.d. increase in GRS. Sample sizes: AFR, 250 cases and 3,977 controls; AMR, $n = 151$ cases and 2,647 controls; EUR, 24,062 cases and 296,594 controls. UKB, UK Biobank. **c**, Odds ratio for COPD per s.d. change in GRS in COPD case–control

studies. $P$ values were calculated from a logistic regression adjusted for age, age squared, sex, height and principal components, followed by fixed-effect meta-analysis. **d**, Decile analysis meta-analyzed across five EUR studies shown as the COPD odds ratio compared between members of each decile and the reference decile. $n = 11,074$ (4,328 cases and 6,746 controls). Statistical tests were two-sided, the height of the bars show the point estimate of the effect and whiskers show the 95% CI. OR, odds ratio.

using sequencing (ATAC–seq) peaks (Supplementary Note) in matrix fibroblast 1 for FVC, matrix fibroblast 2 for $FEV_1$, myofibroblast for $FEV_1$, $FEV_1/FVC$ and PEF, and alveolar type 1 cells in $FEV_1/FVC$; furthermore, genic annotations showed enrichment of exon variants for $FEV_1$, $FEV_1/$ FVC and 3' UTR variants for $FEV_1$ and FVC. We also found enrichment of transcription-factor binding sites in lung across all phenotypes and in bronchus for $FEV_1/FVC$ (Supplementary Table 12).

**Identification of putative causal genes and variants**
To identify putative causal genes, we systematically integrated orthogonal evidence using eight locus- or similarity-based criteria (Supplementary Note): (1) the nearest gene to the sentinel SNP, (2) co-localization

of the GWAS signal and expression quantitative trait loci (eQTL) or (3) protein quantitative trait loci (pQTL) signals in relevant tissues (Methods), (4) rare variant association in whole-exome sequencing in the UK Biobank, (5) proximity to a gene for a Mendelian disease with a respiratory phenotype (±500 kb), (6) proximity to a human ortholog of a mouse-knockout gene with a respiratory phenotype (±500 kb), (7) an annotation-informed credible set[14] containing a missense/del-eterious/damaging variant with a posterior probability of associa-tion >50% and (8) the gene with the highest polygenic priority score (PoPS)[9]. We identified 559 putative causal genes satisfying at least two criteria, of which 135 were supported by at least three criteria (Figs. 1b, 2 and Supplementary Fig. 12). Among the 20 genes supported by

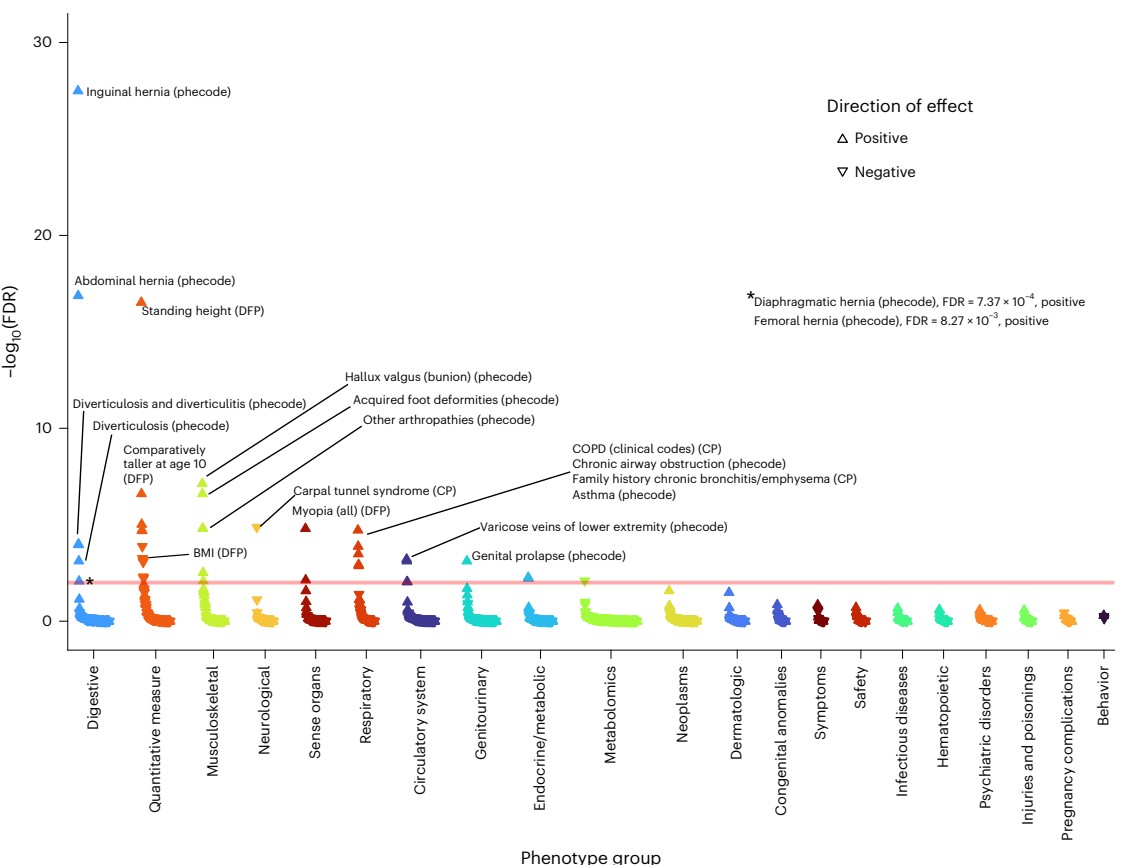

**Fig. 4 | PheWAS for FEV₁/FVC-weighted GRS partitioned according to elastic fiber formation.** Reactome pathway database. CP, composite phenotype and DFP, Data-Field ID phenotype (Methods). The peach-colored line means FDR 1%.

≥4 criteria (Supplementary Table 13), six previously implicated genes (*TGFB2*, *NPNT*, *LTBP4*, *TNS1*, *SMAD3* and *AP3B1*)[5,15,18–20] were supported by additional criteria compared with the original reports. Fourteen of the 20 genes supported by ≥4 criteria have not been previously confidently implicated in lung function (*CYTL1*, *HMCN1*, *GATA5*, *ADAMTS10*, *IGHMBP2*, *SCMH1*, *GLI3*, *ABCA3*, *TIM1*, *CFH*, *FGFR1*, *LRBA*, *CLDN18* and *IGF2BP2*). These are involved in smooth-muscle function (*FGFR1*, *GATA5* and *STIM1*), tissue organization (*ADAMTS10*), alveolar and epithelial function (*ABCA3* and *CLDN18*), and inflammation and immune response to infection (*CFH*, *CYTL1*, *HMCN1*, *LRBA* and *STIM1*).

To supplement our understanding of the biological pathways and clinical phenotypes influenced by lung-function-associated variants, we undertook PheWAS of selected individual variants. We selected 27 putative causal genes implicated by ≥4 criteria (20 genes) or by a single putative causal missense variant that was deleterious (five genes: *ACAN*, *ADGRG6*, *SCARF2*, *CACNA1S* and *HIST1H2BE*) or rare (two genes: *SOS2* and *ADRB2*; Supplementary Table 14). We interpreted the PheWAS findings (shown in full in Supplementary Fig. 13 and Supplementary Table 15) alongside literature reviews (Supplementary Table 16) and highlight examples below.

The putative causal deleterious missense *ABCA3* rs149989682 (A allele; frequency of 0.6%) variant associated with reduced FEV₁/FVC was reported to cause pediatric interstitial lung disease[21]. *ABCA3*, which is expressed in alveolar type II cells and localized to lamellar bodies, is involved in surfactant-phospholipid metabolism, and *ABCA3* mutations cause severe neonatal surfactant deficiency[22]. The putative causal deleterious missense *GATA5* rs200383755 (C allele, frequency of 0.6%) variant associated with lower FEV₁ was associated with increased asthma risk, higher blood pressure and reduced risk of benign prostatic hyperplasia (Supplementary Fig. 13i). *GATA5* associations have not

been previously noted in asthma GWAS, although Gata5-deficient mice show airway hyperresponsiveness[23]. *GATA5* encodes a transcription factor expressed in bronchial smooth muscle, bladder and prostate; a previous benign prostatic hyperplasia GWAS reported a *GATA5* signal[23,24]. *CLDN18* was implicated by four criteria, including a mouse knockout with abnormal pulmonary alveolar epithelium morphology[25]. Through calcium-independent cell adhesion, *CLDN18* influences epithelial-barrier function through tight-junction-specific obliteration of the intercellular space[26]. Its splice variant, CLDN18.1, is predominantly expressed in the lung[27]. Reduced *CLDN18* expression was reported in asthma[26]. However, our PheWAS showed no association with asthma susceptibility or other traits (CLDN18 rs182770 in Supplementary Table 15). *LRBA* was also implicated by four criteria. Mutations resulting in *LRBA* deficiency cause common variable immunodeficiency-8 with autoimmunity, which can include coughing, respiratory infections, bronchiectasis and interstitial lung disease[28,29]. The putative causal *LRBA* tolerated missense variant rs2290846 (posterior probability of 56.3%) was associated with 31 traits (false discovery rate (FDR) < 1%; Supplementary Fig. 13n and Supplementary Table 15); the G allele, associated with lower FVC and lower FEV₁, was associated with lower neutrophils as well as lower risk of cholelithiasis, cholecystitis[30] and diverticular disease.

*FGFR1*, encoding Fibroblast growth factor receptor 1, has roles in lung development and regeneration[31]. Loss-of-function *FGFR1* mutations cause hypogonadotropic hypogonadism[32]. The T allele of rs881299, associated with lower FEV₁/FVC and higher FVC, was strongly associated with higher testosterone (particularly in males) and higher sex-hormone-binding globulin (SHBG), lower body-mass index (BMI) as well as lower levels of alanine transaminase and urate (Supplementary Fig. 13w–y and Supplementary Table 15). The missense *SOS2* variant

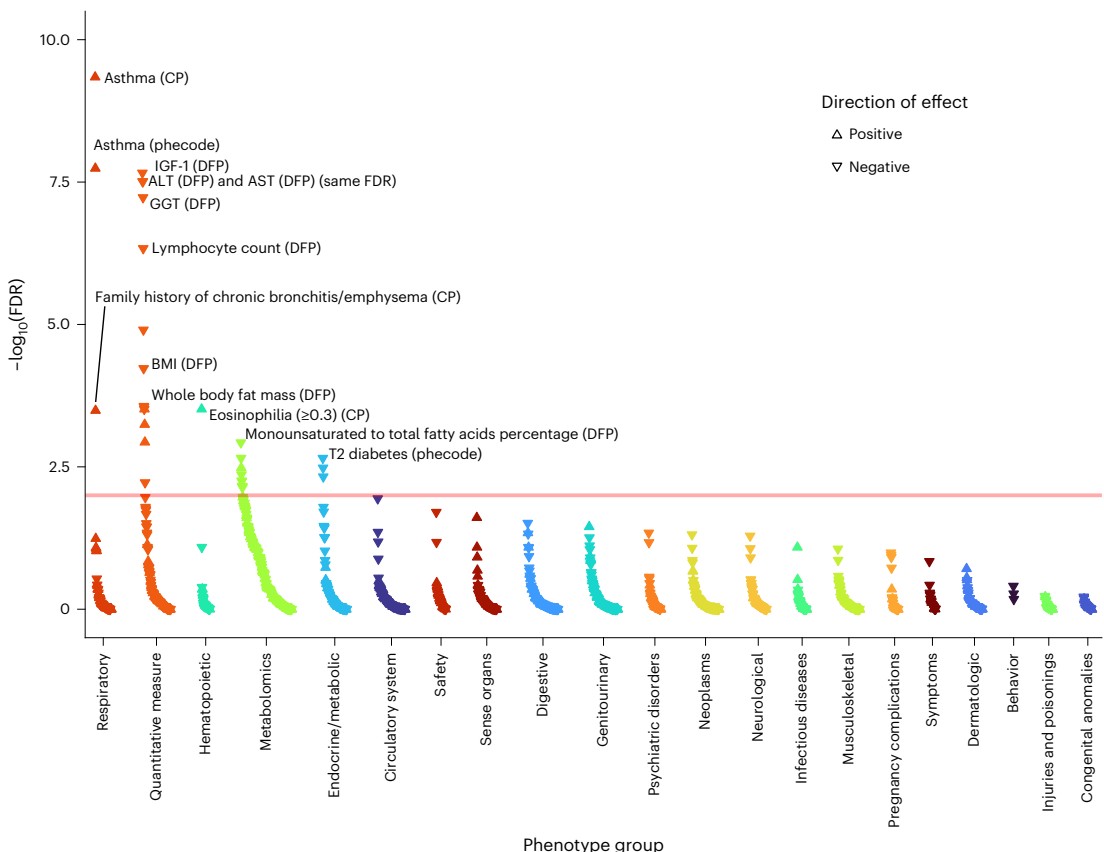

**Fig. 5 | PheWAS for FEV₁/FVC-weighted GRS partitioned according to the PI3K–Akt signaling pathway in *Homo sapiens*.** Kyoto Encyclopedia of Genes and Genomes. CP, composite phenotype; DFP, Data-Field ID phenotype (Methods). The peach-colored line means FDR 1%.

rs72681869 also showed association with SHBG; in both sexes, the G allele, associated with lower FVC and lower FEV₁, was associated with lower SHBG, higher alanine aminotransferase (ALT) and aspartate aminotransferase (AST), higher fat mass, HbA1c and higher systolic and diastolic blood pressure, higher urate and creatinine, and in males lower testosterone and reduced inguinal hernia risk (Supplementary Fig. 13z–bb). Mutations in *SOS2* have been reported in individuals with Noonan syndrome. The A allele of rs7514261 implicating *CFH*, associated with lower FVC, was strongly associated with reduced risk of macular degeneration[33] as well as raised albumin (Supplementary Fig. 13g).

*CACNA1S* is one of several putative causal genes encoding calcium voltage-gated channel subunits in skeletal muscle (*CACNA1S, CACNA1D* and *CACNA2D3* supported by ≥2 criteria; *CACNA1C* was supported by PoPS). *CACNA1S* mutations have been reported in hypokalemic periodic paralysis[34] and malignant hyperthermia[35]. *CACNA1S* is strongly expressed in skeletal muscle but at much lower levels in airway smooth muscle. The common *CACNA1S* missense variant rs3850625 (A allele, frequency of 11.8% in EUR and 21.4% in SAS) was associated with lower FVC, lower FEV₁, lower whole body fat-free mass, reduced hand grip strength as well as lower AST and creatinine levels (Supplementary Fig. 13f). *CACNA1S* and *CACNA1D* are targeted by dihydropyridine calcium channel blockers, which previously produced small improvements in lung function in asthma[36]. For the low-frequency missense *ADRB2* variant rs1800888 (T; 1.49% in EUR), associated with lower FEV₁ and lower FEV₁/FVC, the strongest PheWAS association was with increased eosinophil count (Supplementary Fig. 13d).

**Druggable targets**

Using the Drug Gene Interaction Database, we surveyed 559 genes supported by ≥2 criteria. CheMBL interactions identified 292 drugs mapping to 55 genes (Supplementary Table 17), including *ITGA2*, which encodes

integrin subunit alpha 2. The reduced expression of *ITGA2* in lung tissue associated with the C allele of rs12522114 mimics vatelizumab-induced ITGA2 inhibition; this allele is associated with higher FEV₁ and FEV₁/FVC, indicating the potential to repurpose vatelizumab, which increases T regulatory cell populations[37], for COPD treatment.

**Pathway analysis**

Using ConsensusPathDB[38], we tested biological pathway enrichment for 559 causal genes supported by ≥2 criteria, highlighting pathways relevant for development, tissue integrity and remodeling (Supplementary Table 18). These include pathways not previously implicated in pathway enrichment analyses for lung function—such as PI3K–Akt signaling, integrin pathways, endochondral ossification, calcium signaling, hypertrophic cardiomyopathy and dilated cardiomyopathy—as well as those previously implicated via individual genes[5] such as TNF signaling, actin cytoskeleton, AGE–RAGE signaling, Hedgehog signaling and cancers. We found strengthened enrichment through newly identified genes in previously described pathways, such as extracellular matrix organization (34 new genes), elastic fiber formation (eight new genes) and TGF–Core (four new genes). Consistent with our ConsensusPathDB findings, Ingenuity Pathway Analysis (https://digitalinsights.qiagen.com/IPA)[39] highlighted enrichment of cardiac hypertrophy signaling and osteoarthritis pathways and also implicated pulmonary and hepatic fibrosis signaling pathways, axonal guidance and PTEN signaling as well as the upstream regulators TGFB1 and IGF-1 (Supplementary Table 19).

**Multi-ancestry GRS for FEV1/FVC and COPD**

We built multi-ancestry and ancestry-specific GRSs weighted by FEV₁/FVC effect sizes and tested association with FEV₁/FVC and COPD (GOLD stage 2–4) within groups of individuals of different ancestries in the UK Biobank (Methods). Our new GRS improved lung-function and

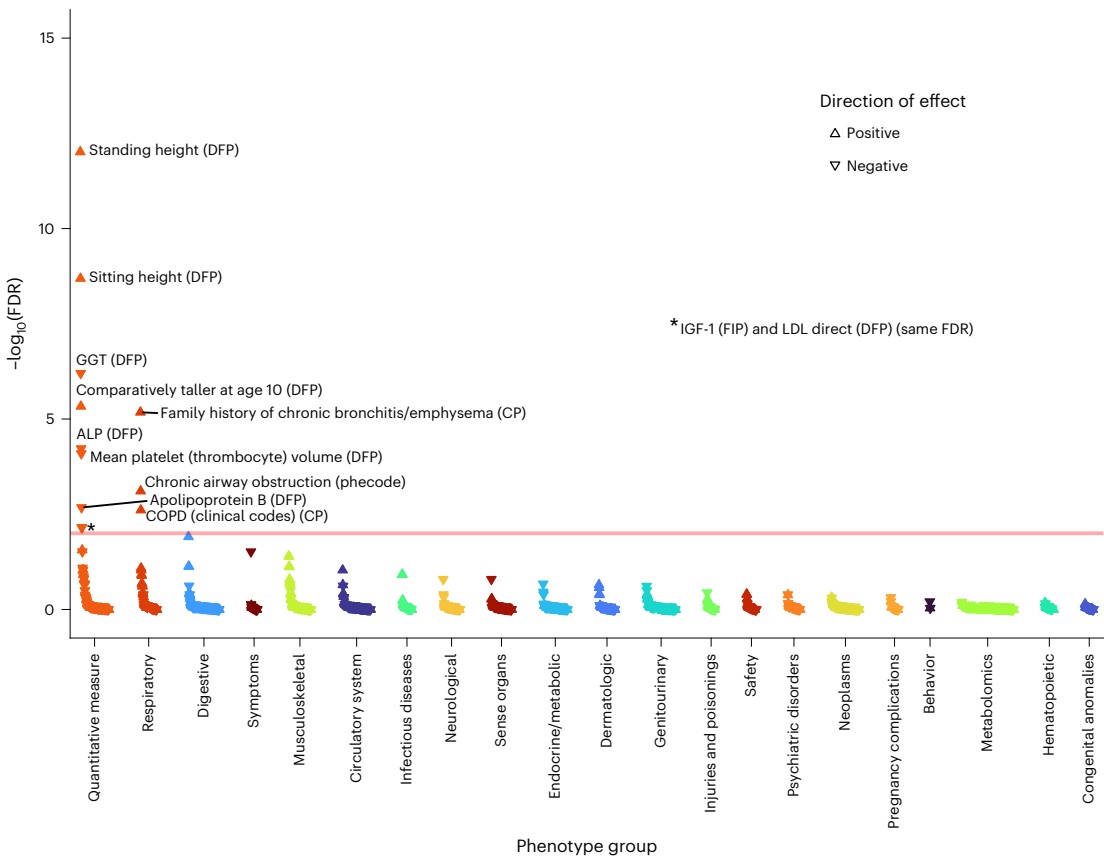

**Fig. 6 | PheWAS for FEV$_1$/FVC-weighted GRS partitioned according to hypertrophic cardiomyopathy in *H. sapiens*.** Kyoto Encyclopedia of Genes and Genomes. CP, composite phenotype; DFP, Data-Field ID phenotype (Methods). The peach-colored line means FDR 1%.

COPD prediction compared with a previous GRS based only on individuals of EUR ancestry[5] (Fig. 3a,b and Supplementary Table 20), and the multi-ancestry GRS outperformed the ancestry-specific GRS in all UK Biobank ancestries. We then tested the multi-ancestry GRS in five independent COPD case–control studies (Supplementary Table 21 and Methods). Stronger COPD susceptibility associations were observed across five EUR-ancestry studies compared with a previous GRS[5] (Fig. 3c and Supplementary Table 22). In the meta-analysis of these EUR studies, the odds ratio for COPD per s.d. of GRS increase was 1.63 (95% confidence interval (CI), 1.56–1.71; $P = 7.1 \times 10^{-93}$); members of the highest GRS decile had a 5.16-fold higher COPD risk than the lowest decile (95% CI, 4.14–6.42; $P = 1.0 \times 10^{-48}$; Fig. 3d and Supplementary Table 23). The results for individuals in the SPIROMICS study of AFR ancestry were comparable to individuals from the UK Biobank with AFR ancestry but lower in magnitude compared with the COPDGene AFR population (Fig. 3c).

### PheWAS of trait-specific GRSs
To study the aggregate effects of lung-function-associated genetic variants on a wide range of diseases and disease-relevant traits, we created GRSs for FEV$_1$, FVC, FEV$_1$/FVC and PEF, each comprising sentinel variants ($P < 5 \times 10^{-9}$) with weights estimated from the multi-ancestry meta-regression (Methods), and tested these in PheWAS. These GRS values showed distinct patterns of associations with respiratory and non-respiratory phenotypes (Supplementary Fig. 14 and Supplementary Table 24). A GRS for lower FEV$_1$ was most strongly associated with increased risk of asthma and COPD, family history of chronic bronchitis/emphysema, lower hand grip strength, increased fat mass, increased HbA1c and type 2 diabetes risk, and elevated C-reactive protein. In addition, associations were observed with increased asthma exacerbations

and lower age of onset for COPD (Supplementary Fig. 14a). The GRS for lower FEV$_1$/FVC was associated with key respiratory phenotypes: increased risk of COPD and asthma, family history of chronic bronchitis/emphysema, increased emphysema risk, increased risk of respiratory insufficiency or respiratory failure and younger age of onset for COPD but a slightly lower risk of COPD exacerbations (Supplementary Fig. 14b). In contrast, the GRS for lower FVC was strongly associated with many traits—among the strongest associations were high C-reactive protein, increased fat mass, raised HbA1c and type 2 diabetes, raised systolic blood pressure, lower hand grip strength and raised ALT as well as increased risk of clinical codes for asthma and COPD (Supplementary Fig. 14c). Although the GRS for lower FEV$_1$/FVC was associated with increased standing and sitting height, the GRSs for lower FEV$_1$ and FVC were associated with increased standing height but reduced sitting height. Broadly similar phenome-wide associations were seen for the PEF and the FEV$_1$ GRS (Supplementary Fig. 14d).

### PheWAS of GRSs partitioned by pathway
Finally, we hypothesized that partitioning our lung-function GRS into pathway-specific GRSs according to the biological pathways the variants influence could inform understanding of mechanisms underlying impaired lung function, and the probable consequences of perturbing specific pathways. Informed by the above prioritization of putative causal genes and classification of these genes by pathway ('Pathway analysis' section), we undertook PheWAS for FEV$_1$/FVC-weighted GRSs partitioned by each of the 29 pathways enriched (FDR < 10$^{-5}$) for the 559 genes implicated by ≥2 criteria (Methods). Partitioning of GRSs in this way highlighted markedly different patterns of phenome-wide associations (Supplementary Fig. 15 and Supplementary Table 25). Figures 4–7 highlight four pathway-specific GRS examples; all demonstrated

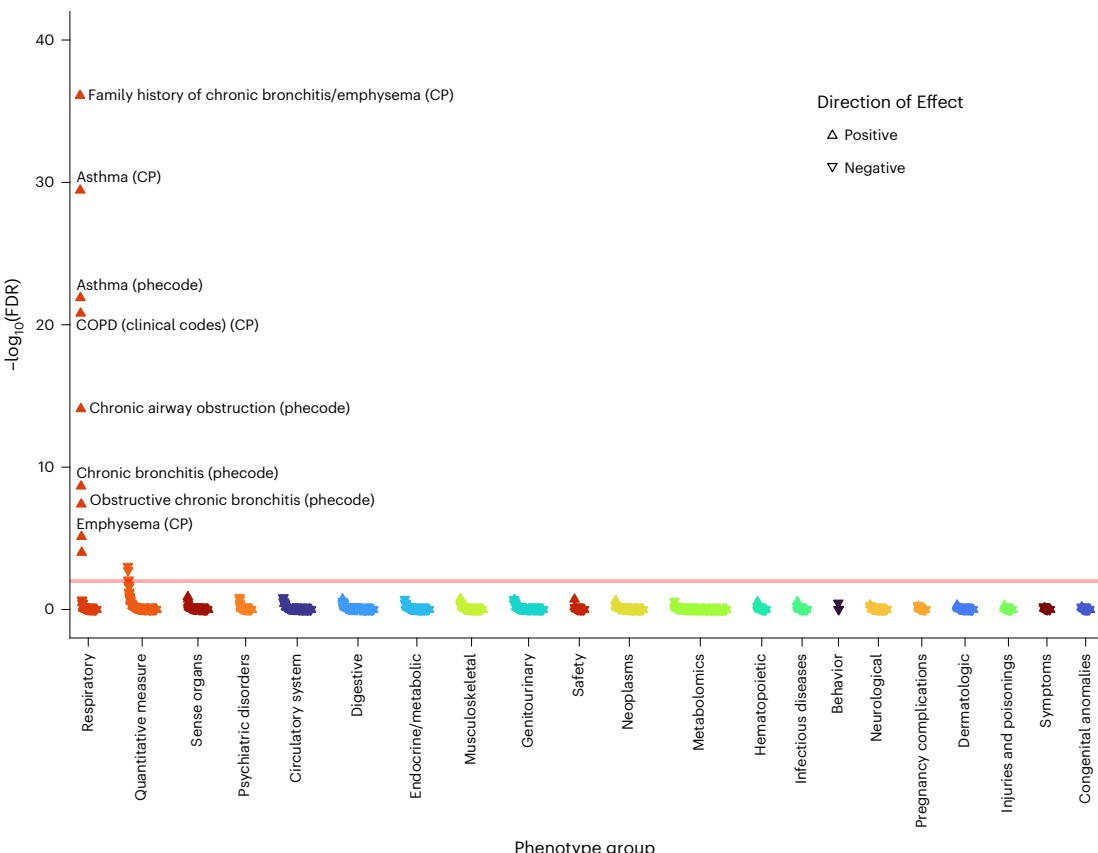

**Fig. 7 | PheWAS for FEV$_1$/FVC-weighted GRS partitioned according to signal transduction.** Reactome pathway database. CP, composite phenotype (Methods). The peach-colored line means FDR 1%.

association with COPD clinical codes and family history of chronic bronchitis/emphysema, although the associations with other traits varied. The GRS for lower FEV$_1$/FVC specific to elastic fiber formation was associated with increased risk of inguinal, abdominal, diaphragmatic and femoral hernia; diverticulosis; arthropathies; hallux valgus as well as genital prolapse; reduced carpal tunnel syndrome risk and BMI; and increased asthma risk (Fig. 4). In contrast, the GRS for lower FEV$_1$/FVC specific to PI3K–Akt signaling was associated with increased asthma risk, lower IGF-1, lower liver enzymes (ALT, AST and gamma glutamyltransferase (GGT)), lower lymphocyte counts, raised eosinophils, lower fat mass and BMI, and reduced diabetes risk (Fig. 5). The GRS for lower FEV$_1$/FVC specific to the hypertrophic cardiomyopathy pathway was associated with reduced liver enzymes (ALT and GGT) as well as lower apolipoprotein B, LDL, IGF-1 and mean platelet volume (Fig. 6). The GRS associations for lower FEV$_1$/FVC partitioned to signal transduction were specific to respiratory traits, including asthma and emphysema (Fig. 7). Variable height associations were evident: the GRS for lower FEV$_1$/FVC showed association with increased height when partitioned to elastic fiber formation or hypertrophic cardiomyopathy (Figs. 4 and 6), reduced height when partitioned to ESC pluripotency (Supplementary Fig. 15g) and no height association when partitioned to PI3K–Akt signaling or signal transduction (Figs. 5 and 7).

We hypothesized that individuals may have high GRS for ≥1 pathways and low GRS for other pathways. Comparisons of the GRSs of individuals across pairs of pathways for each of the 29 pathways (Supplementary Fig. 16a) and in detail for the elastic fiber, PI3K–Akt signaling, hypertrophic cardiomyopathy and signal transduction pathways (Supplementary Fig. 16b) demonstrated how GRS profiles may be concordant or discordant across pathways, which could have implications for the choice of therapy.

## Discussion

We present a large ancestrally diverse lung-function GWAS and a comprehensive initiative to relate lung-function- and COPD-associated variants to functional annotations, cell types, genes and pathways. It is the first to investigate possible consequences of intervening in relevant pathways through PheWAS studies, utilizing pathway-partitioned GRS.

The 1,020 signals identified were enriched in functionally active regions in alveolar type 1 cells, fibroblasts, myofibroblasts, bronchial epithelial cells, and adult and fetal lung. We showed effect heterogeneity attributable to ancestry for 109 signals (including *LTBP4*, *THSD4*, *EFEMP1* and *MECOM*), between ever-smokers and never-smokers (*HTR4*), and differences in effects between adults and children (including *CCDC91* and *NPNT*). We mapped lung-function signals to 559 putatively causal genes meeting ≥2 independent criteria. Exemplar genes supported by ≥4 criteria or by deleterious or rare putative causal missense variants implicated surfactant-phospholipid metabolism, smooth-muscle function, epithelial morphology and barrier function, innate immunity, calcium signaling, adrenoceptor signaling, and lung development and regeneration. Among the pathways enriched for putative causal genes were PI3K–Akt signaling, integrin pathways, endochondral ossification, calcium signaling, hypertrophic cardiomyopathy and dilated cardiomyopathy. These pathways have not been previously implicated in lung function using GWAS.

Combined as a GRS weighted by FEV$_1$/FVC effect size, the 1,020 variants strongly predicted COPD in the UK Biobank and in COPD case–control studies, with a more than fivefold change in risk between the highest and lowest GRS deciles. This GRS more strongly predicted FEV$_1$/FVC and COPD across all ancestries than a previous GRS[5]. Partitioning the FEV$_1$/FVC GRS by the pathways defined by specific variants, informed by detailed, systematic variant-to-gene mapping and

pathway analyses, and using our new Deep-PheWAS platform[40], illustrated unique patterns of phenotype associations for each pathway GRS. These patterns of PheWAS findings are relevant to the potential efficacy and side effects of intervention in these pathways. As a proof-of-concept, the GRS associated with lower $FEV_1/FVC$ specific to PI3K–Akt signaling was associated with increased risk of COPD but a lower risk of diabetes; PI3K inhibition impairs glucose uptake in muscle and increases hepatic gluconeogenesis, contributing to glucose intolerance and diabetes[41]. The PheWAS and druggability analyses we conducted have the potential to identify drug repurposing opportunities for COPD.

The patterns of pleiotropy we show through PheWAS for individual variants, trait-specific GRS and pathway-partitioned GRS may help explain variants and pathways that increase susceptibility to more than one disease and thereby predispose to particular patterns of multimorbidity. For example, the elastic fiber pathway GRS was associated with increased risk of muscular (for example, hernias) and musculoskeletal conditions related to connective-tissue laxity. Our findings also further inform the complex relationship between height, BMI and obesity, and lung function and their genetic determinants[5,42]. Lung-function and height associations were uncorrelated, and height relationships differed between GRS for different lung-function traits, and even between sitting and standing height for the same trait. The pathway-partitioned GRS analyses indicate that the relationship between genetic variants, height and lung-function traits depends on the pathways through which the variants act.

The last comprehensive attempt to map lung-function-associated variants to genes identified 107 putative causal genes, mostly through eQTLs only, and only eight genes were then implicated by ≥2 criteria[5]. In contrast, we implicated 559 causal genes meeting ≥2 criteria by drawing on new data and methodologies, such as single-cell epigenome data, rare variant associations identified in sequencing data in the UK Biobank and similarity-based approach PoPS[9]. Nevertheless, our study has limitations. We focused on multi-ancestry rather than ancestry-specific signals, as the sample sizes for lung-function genomics studies in all non-EUR ancestry groups were limited, particularly for the AFR ancestries[4]. Non-EUR ancestries are under-represented in genomic studies[3], constraining GWAS and PheWAS studies in these populations. Correcting this will require substantial global investment in suitably phenotyped and genotyped studies, with appropriate community participation and workforce development. Improved sample sizes across all ancestries would improve power in ancestry-specific studies[42] and fine mapping of multi-ancestry meta-analysis signals.

Strategies for in silico mapping of association signals to causal genes are evolving and difficult to evaluate without a reference set of fully functionally characterized lung-function-associated variants and causal genes. Our variant-to-gene mapping framework parallels one that was recently adopted[10] and could help prioritization of genes for functional experiments such as gene editing in relevant organoids with appropriate readouts to confirm mechanism. An additional limitation is that classifications of pathways may be imperfect; we used multiple pathway classifications as it is unclear which is superior across all component pathways and we present the pathway-partitioned PheWAS results as a resource to others.

In summary, our multi-ancestry study highlights new putative causal variants, genes and pathways, some of which are targeted by existing drug compounds. These findings bring us closer to understanding mechanisms underlying lung function and COPD and will inform functional genomics experiments to confirm mechanisms and consequently guide the development of therapies for impaired lung function and COPD.

## Online content

Any methods, additional references, Nature Portfolio reporting summaries, source data, extended data, supplementary information,

acknowledgements, peer review information; details of author contributions and competing interests; and statements of data and code availability are available at https://doi.org/10.1038/s41588-023-01314-0.

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

**Nick Shrine** [1,135] ✉, **Abril G. Izquierdo**[1,135], **Jing Chen** [1,135], **Richard Packer**[1,135], **Robert J. Hall**[2], **Anna L. Guyatt** [1], **Chiara Batini**[1,3], **Rebecca J. Thompson**[2], **Chandan Pavuluri**[4], **Vidhi Malik**[4], **Brian D. Hobbs** [4,5], **Matthew Moll**[4], **Wonji Kim** [4], **Ruth Tal-Singer**[6], **Per Bakke**[7], **Katherine A. Fawcett**[1], **Catherine John**[1,3], **Kayesha Coley** [1], **Noemi Nicole Piga**[1], **Alfred Pozarickij**[8], **Kuang Lin**[8], **Iona Y. Millwood**[8,9], **Zhengming Chen** [8,9], **Liming Li** [10], **China Kadoorie Biobank Collaborative Group**[*], **Sara R. A. Wijnant**[11,12,13], **Lies Lahousse** [12,13], **Guy Brusselle**[11,13], **Andre G. Uitterlinden** [14], **Ani Manichaikul**[15], **Elizabeth C. Oelsner** [16], **Stephen S. Rich** [15], **R. Graham Barr**[16], **Shona M. Kerr** [17], **Veronique Vitart** [17], **Michael R. Brown**[18], **Matthias Wielscher**[19], **Medea Imboden**[20,21], **Ayoung Jeong** [20,21], **Traci M. Bartz**[22], **Sina A. Gharib** [23], **Claudia Flexeder**[24,25,26], **Stefan Karrasch**[24,25,26], **Christian Gieger** [26,27], **Annette Peters** [26,28], **Beate Stubbe**[29], **Xiaowei Hu**[15], **Victor E. Ortega**[30], **Deborah A. Meyers**[31], **Eugene R. Bleecker**[31], **Stacey B. Gabriel**[32], **Namrata Gupta**[32], **Albert Vernon Smith** [33,34], **Jian'an Luan** [35], **Jing-Hua Zhao**[36], **Ailin F. Hansen**[37], **Arnulf Langhammer** [38,39], **Cristen Willer** [40,41,42], **Laxmi Bhatta**[37], **David Porteous** [43], **Blair H. Smith** [44], **Archie Campbell** [43], **Tamar Sofer** [45,46,47], **Jiwon Lee** [45], **Martha L. Daviglus**[48], **Bing Yu**[49], **Elise Lim** [50], **Hanfei Xu**[50],

George T. O'Connor[51], Gaurav Thareja [52], Omar M. E. Albagha [53,54], The Qatar Genome Program Research (QGPR) Consortium\*, Karsten Suhre [52,56], Raquel Granell [57], Tariq O. Faquih [58], Pieter S. Hiemstra [59], Annelies M. Slats[59], Benjamin H. Mullin[60,61], Jennie Hui[62,63,64], Alan James[62], John Beilby[61,62], Karina Patasova[65,66], Pirro Hysi [65,67], Jukka T. Koskela [68], Annah B. Wyss[69], Jianping Jin[70], Sinjini Sikdar [69,71], Mikyeong Lee [69], Sebastian May-Wilson [72], Nicola Pirastu[72], Katherine A. Kentistou[72,73], Peter K. Joshi [72], Paul R. H. J. Timmers [72], Alexander T. Williams[1], Robert C. Free[3,74], Xueyang Wang[3,74], John L. Morrison[75], Frank D. Gilliland [75], Zhanghua Chen[75], Carol A. Wang [76,77], Rachel E. Foong[78,79], Sarah E. Harris [80], Adele Taylor[80], Paul Redmond[80], James P. Cook[81], Anubha Mahajan [82,83], Lars Lind[84], Teemu Palviainen [85], Terho Lehtimäki[86], Olli T. Raitakari[87,88], Jaakko Kaprio [85], Taina Rantanen[89], Kirsi H. Pietiläinen [90,91], Simon R. Cox [80], Craig E. Pennell [76,77,92], Graham L. Hall[78,79], W. James Gauderman[75], Chris Brightling[3,93], James F. Wilson [72,94], Tuula Vasankari [95,96], Tarja Laitinen[97], Veikko Salomaa [98], Dennis O. Mook-Kanamori[58,99], Nicholas J. Timpson [57,100], Eleftheria Zeggini[55,101,102], Josée Dupuis [103], Caroline Hayward[17], Ben Brumpton [38,104], Claudia Langenberg [35,105,106], Stefan Weiss [107], Georg Homuth [107], Carsten Oliver Schmidt[108], Nicole Probst-Hensch[20,21], Marjo-Riitta Jarvelin [19,109,110,111], Alanna C. Morrison [18], Ozren Polasek[112], Igor Rudan [113], Joo-Hyeon Lee [114,115], Ian Sayers [2], Emma L. Rawlins [116], Frank Dudbridge [1], Edwin K. Silverman[4], David P. Strachan [117], Robin G. Walters [8,9], Andrew P. Morris [118], Stephanie J. London [69], Michael H. Cho [4], Louise V. Wain [1,3], Ian P. Hall [2,135] & Martin D. Tobin [1,3,135] ✉

[1]Department of Population Health Sciences, University of Leicester, Leicester, UK. [2]Division of Respiratory Medicine and NIHR Nottingham Biomedical Research Centre, University of Nottingham, Nottingham, UK. [3]Leicester National Institute for Health and Care Research, Biomedical Research Centre, Glenfield Hospital, Leicester, UK. [4]Channing Division of Network Medicine, Division of Pulmonary and Critical Care Medicine, Department of Medicine, Brigham and Women's Hospital, Boston, MA, USA. [5]Harvard Medical School, Boston, MA, USA. [6]COPD Foundation, Washington DC, USA. [7]Department of Clinical Science, Unversity of Bergen, Bergen, Norway. [8]Nuffield Department of Population Health, University of Oxford, Oxford, UK. [9]MRC Population Health Research Unit, University of Oxford, Oxford, UK. [10]Department of Epidemiology and Biostatistics, School of Public Health, Peking University Health Science Center, Beijing, China. [11]Department of Respiratory Diseases, Ghent Universital Hospital, Ghent, Belgium. [12]Department of Bioanalysis, Faculty of Pharmaceutical Sciences, Ghent University, Ghent, Belgium. [13]Department of Epidemiology, Eramus Medical Center, Rotterdam, The Netherlands. [14]Department of Internal Medicine, Eramus Medical Center, Rotterdam, The Netherlands. [15]Center for Public Health Genomics, University of Virginia, Charlottesville, VA, USA. [16]Department of Medicine, Columbia University Medical Center, New York, NY, USA. [17]Medical Research Council Human Genetics Unit, Institute of Genetics and Cancer, University of Edinburgh, Edinburgh, UK. [18]Human Genetics Center, Department of Epidemiology, Human Genetics, and Environmental Sciences, School of Public Health, The University of Texas Health Science Center at Houston, Houston, TX, USA. [19]MRC Centre for Environment and Health, Department of Epidemiology and Biostatistics, School of Public Health, Imperial College London, London, UK. [20]Department of Epidemiology and Public Health, Swiss Tropical and Public Health Institute, Allschwil, Switzerland. [21]Department of Public Health, University of Basel, Basel, Switzerland. [22]Cardiovascular Health Research Unit, Departments of Medicine and Biostatistics, University of Washington, Seattle, WA, USA. [23]Computational Medicine Core, Center for Lung Biology, Division of Pulmonary, Critical Care and Sleep Medicine, Department of Medicine, University of Washington, Seattle, WA, USA. [24]Institute and Clinic for Occupational, Social and Environmental Medicine, University Hospital, LMU Munich, Munich, Germany. [25]Comprehensive Pneumology Center Munich (CPC-M), German Center for Lung Research (DZL), Munich, Germany. [26]Institute of Epidemiology, Helmholtz Zentrum München–German Research Center for Environmental Health, Neuherberg, Germany. [27]Research Unit of Molecular Epidemiology, Helmholtz Zentrum München–German Research Center for Environmental Health, Neuherberg, Germany. [28]Institute for Medical Information Processing, Biometry and Epidemiology, Medical Faculty, Ludwig Maximilian University, Munich, Germany. [29]Department of Internal Medicine B–Cardiology, Intensive Care, Pulmonary Medicine and Infectious Diseases, University Medicine Greifswald, Greifswald, Germany. [30]Division of Respiratory Medicine, Department of Internal Medicine, Center for Individualized Medicine, Mayo Clinic, Scottsdale, AZ, USA. [31]Department of Medicine, University of Arizona, Tucson, AZ, USA. [32]Broad Institute of MIT and Harvard, Cambridge, MA, USA. [33]Department of Biostatistics, University of Michigan School of Public Health, Ann Arbor, MI, USA. [34]Center for Statistical Genetics, University of Michigan School of Public Health, Ann Arbor, MI, USA. [35]MRC Epidemiology Unit, Institute of Metabolic Science, School of Clinical Medicine, University of Cambridge, Cambridge, UK. [36]Department of Public and Primary Care, Heart and Lung Research Institute, University of Cambridge, Cambridge, UK. [37]K.G. Jebsen Center for Genetic Epidemiology, Department of Public Health and Nursing, NTNU Norwegian University of Science and Technology, Trondheim, Norway. [38]HUNT Research Centre, Department of Public Health and Nursing, NTNU Norwegian University of Science and Technology, Levanger, Norway. [39]Levanger Hospital, Nord-Trøndelag Hospital Trust, Levanger, Norway. [40]Division of Cardiology, Department of Internal Medicine, University of Michigan, Ann Arbor, MI, USA. [41]Department of Biostatistics and Center for Statistical Genetics, University of Michigan, Ann Arbor, MI, USA. [42]Department of Human Genetics, University of Michigan, Ann Arbor, MI, USA. [43]Centre for Genomic and Experimental Medicine, Institute of Genetics and Cancer, University of Edinburgh, Edinburgh, UK. [44]Division of Population Health and Genomics, Ninewells Hospital and Medical School, University of Dundee, Dundee, UK. [45]Division of Sleep and Circadian Disorders, Brigham and Women's Hospital, Boston, MA, USA. [46]Department of Medicine, Harvard Medical School, Boston, MA, USA. [47]Department of Biostatistics, Harvard T.H. Chan School of Public Health, Boston, MA, USA. [48]Institute for Minority Health Research, University of Illinois at Chicago, Chicago, IL, USA. [49]Human Genetics Center, Department of Epidemiology, Human Genetics and Environmental Sciences, School of Public Health, University of Texas Health Science Center at Houston, Houston, TX, USA. [50]Department of Biostatistics, School of Public Health, Boston University, Boston, MA, USA. [51]Pulmonary Center, School of Medicine, Boston University, Boston, MA, USA. [52]Bioinformatics Core, Weill Cornell Medicine–Qatar, Education City, Doha, Qatar. [53]College of Health and Life Sciences, Hamad Bin Khalifa University, Doha, Qatar. [54]Center for Genomic and Experimental Medicine, Institute of Genetics and Cancer, University of Edinburgh, Edinburgh, UK. [55]Wellcome Sanger Institute, Cambridge, UK. [56]Department of Biophysics and Physiology, Weill Cornell Medicine, New York, NY, USA. [57]MRC Integrative Epidemiology Unit (IEU), Population Health Sciences, Bristol Medical School, University of Bristol, Bristol, UK. [58]Department of

Clinical Epidemiology, Leiden University Medical Center, Leiden, The Netherlands. [59]Department of Pulmonology, Leiden University Medical Center, Leiden, The Netherlands. [60]Department of Endocrinology and Diabetes, Sir Charles Gairdner Hospital, Nedlands, Western Australia, Australia. [61]School of Biomedical Sciences, University of Western Australia, Crawley, Western Australia, Australia. [62]Busselton Population Medical Research Institute, QEII Medical Centre, Nedlands, Western Australia, Australia. [63]School of Population and Global Health, University of Western Australia, Crawley, Western Australia, Australia. [64]PathWest Laboratory Medicine of WA, Nedlands, Western Australia, Australia. [65]Department of Twin Research and Genetic Epidemiology, King's College London School of Medicine, London, UK. [66]Division of Respiratory Medicine, Department of Medicine Solna, Karolinska Institutet, Karolinska University Hospital, Stockholm, Sweden. [67]UCL Institute of Ophthalmology, University College London, London, UK. [68]Institute for Molecular Medicine Finland (FIMM), University of Helsinki, Helsinki, Finland. [69]Epidemiology Branch, National Institute of Environmental Health Sciences, National Institutes of Health, Department of Health and Human Services, Research Triangle Park, NC, USA. [70]Westat, Durham, NC, USA. [71]Department of Mathematics and Statistics, Old Dominion University, Norfolk, VA, USA. [72]Centre for Global Health Research, Usher Institute for Population Health Sciences and Informatics, University of Edinburgh, Edinburgh, UK. [73]Centre for Cardiovascular Sciences, Queen's Medical Research Institute, University of Edinburgh, Edinburgh, UK. [74]Department of Respiratory Sciences, University of Leicester, Leicester, UK. [75]Department of Population and Public Health Sciences, Keck School of Medicine, University of Southern California, Los Angeles, CA, USA. [76]School of Medicine and Public Health, College of Health, Medicine and Wellbeing, University of Newcastle, Newcastle, New South Wales, Australia. [77]Hunter Medical Research Institute, Newcastle, New South Wales, Australia. [78]Wal-yan Respiratory Research Centre, Telethon Kids Institute, Perth, Western Australia, Australia. [79]School of Allied Health, Faculty of Health Sciences, Curtin University, Perth, Western Australia, Australia. [80]Lothian Birth Cohorts group, Department of Psychology, University of Edinburgh, Edinburgh, UK. [81]Department of Health Data Science, University of Liverpool, Liverpool, UK. [82]Wellcome Centre for Human Genetics, University of Oxford, Oxford, UK. [83]Genentech, South San Francisco, CA, USA. [84]Department of Medical Sciences, Uppsala University, Uppsala, Sweden. [85]Institute for Molecular Medicine Finland–FIMM, University of Helsinki, Helsinki, Finland. [86]Department of Clinical Chemistry, Fimlab Laboratories, and Finnish Cardiovascular Research Center–Tampere, Faculty of Medicine and Health Technology, Tampere University, Tampere, Finland. [87]Department of Clinical Physiology and Nuclear Medicine, Turku University Hospital, Turku, Finland. [88]Research Centre of Applied and Preventive Cardiovascular Medicine, University of Turku, Turku, Finland. [89]Faculty of Sport and Health Sciences, University of Jyvaskyla, Jyvaskyla, Finland. [90]Obesity Research Unit, Research Program for Clinical and Molecular Metabolism, Faculty of Medicine, University of Helsinki, Helsinki, Finland. [91]Obesity and Abdominal Centers, Helsinki University Hospital and University of Helsinki, Helsinki, Finland. [92]Department of Maternity and Gynaecology, John Hunter Hospital, Newcastle, New South Wales, Australia. [93]Department of Infection, Inflammation and Immunity, Institute for Lung Health, University of Leicester, Leicester, UK. [94]MRC Human Genetics Unit, Institute of Genetics and Cancer, University of Edinburgh, Western General Hospital, Edinburgh, UK. [95]FILHA–Finnish Lung Health Association, Helsinki, Finland. [96]Department of Respiratory Diseases and Allergology, University of Turku, Turku, Finland. [97]Administration Center, Tampere University Hospital and University of Tampere, Tampere, Finland. [98]Department of Public Health and Welfare, Finnish Institute for Health and Welfare, Helsinki, Finland. [99]Department of Public Health and Primary Care, Leiden University Medical Center, Leiden, The Netherlands. [100]ALSPAC, Department of Population Health Sciences, Bristol Medical School, University of Bristol, Bristol, UK. [101]Institute of Translational Genomics, Helmholtz Zentrum München–German Research Center for Environmental Health, Neuherberg, Germany. [102]Technical University of Munich (TUM) and Klinikum Rechts der Isar, TUM School of Medicine, Munich, Germany. [103]Department of Epidemiology, Biostatistics, and Occupational Health, School of Population and Global Health, McGill University, Montreal, Quebec, Canada. [104]Clinic of Medicine, St. Olavs Hospital, Trondheim University Hospital, Trondheim, Norway. [105]Precision Healthcare University Research Institute, Queen Mary University of London, London, UK. [106]Computational Medicine, Berlin Institute of Health at Charité, Universitätsmedizin Berlin, Berlin, Germany. [107]Interfaculty Institute for Genetics and Functional Genomics, Department of Functional Genomics, University Medicine Greifswald, Greifswald, Germany. [108]Institute for Community Medicine, SHIP–Clinical Epidemiological Research, University Medicine Greifswald, Greifswald, Germany. [109]Center for Life Course Health Research, Faculty of Medicine, University of Oulu, Oulu, Finland. [110]Biocenter Oulu, University of Oulu, Oulu, Finland. [111]Unit of Primary Health Care, Oulu University Hospital, OYS, Oulu, Finland. [112]School of Medicine, University of Split, Split, Croatia. [113]Centre for Global Health, Usher Institute, University of Edinburgh, Edinburgh, UK. [114]Jeffrey Cheah Biomedical Centre, Wellcome–MRC Cambridge Stem Cell Institute, University of Cambridge, Cambridge, UK. [115]Department of Physiology, Development and Neuroscience, University of Cambridge, Cambridge, UK. [116]Wellcome Trust–CRUK Gurdon Institute and Department of Physiology, Development and Neuroscience, University of Cambridge, Cambridge, UK. [117]Population Health Research Institute, St George's University of London, London, UK. [118]Centre for Genetics and Genomics Versus Arthritis, Division of Musculoskeletal and Dermatological Sciences, Centre for Musculoskeletal Research, The University of Manchester, Manchester, UK. [135]These authors contributed equally: Nick Shrine, Abril G. Izquierdo, Jing Chen, Richard Packer, Ian P. Hall, Martin D. Tobin. *Lists of members and their affiliations appear at the end of the paper. ✉e-mail: nick.shrine@leicester.ac.uk; martin.tobin@leicester.ac.uk

## China Kadoorie Biobank Collaborative Group

Alfred Pozarickij[8], Kuang Lin[8], Iona Y. Millwood[8,9], Zhengming Chen[8,9], Liming Li[10] & Robin G. Walters[8,9]

A full list of members and their affiliations appears in the Supplementary Information.

## The Qatar Genome Program Research (QGPR) Consortium

Said I. Ismail[119], Wadha Al-Muftah[119], Radja Badji[119], Hamdi Mbarek[119], Dima Darwish[119], Tasnim Fadl[119], Heba Yasin[119], Maryem Ennaifar[119], Rania Abdellatif[119], Fatima Alkuwari[119], Muhammad Alvi[119], Yasser Al-Sarraj[119], Chadi Saad[119] & Asmaa Althani[119,120]

[119]Qatar Genome Program, Qatar Foundation Research Development and Innovation, Qatar Foundation, Doha, Qatar. [120]Qatar Biobank for Medical Research, Qatar Foundation, Doha, Qatar.

## Biobank and Sample Preparation

Eleni Fethnou[120], Fatima Qafoud[120], Eiman Alkhayat[120] & Nahla Afifi[120]

## Sequencing and Genotyping group

**Sara Tomei[121], Wei Liu[121] & Stephan Lorenz[121]**

[121]Integrated Genomics Services, Sidra Medicine, Doha, Qatar.

## Applied Bioinformatics Core

**Najeeb Syed[122], Hakeem Almabrazi[122], Fazulur Rehaman Vempalli[122] & Ramzi Temanni[122]**

[122]Applied Bioinformatics Core, Sidra Medicine, Doha, Qatar.

## Data Management and Computing Infrastructure group

**Tariq Abu Saqri[123], Mohammedhusen Khatib[123], Mehshad Hamza[123], Tariq Abu Zaid[123], Ahmed El Khouly[123], Tushar Pathare[123], Shafeeq Poolat[123] & Rashid Al-Ali[123]**

[123]Biomedical Informatics, Sidra Medicine, Doha, Qatar.

## Consortium Lead Principal Investigators

**Omar M. E. Albagha[53,54], Souhaila Al-Khodor[124], Mashael Alshafai[125], Ramin Badii[126], Lotfi Chouchane[127], Xavier Estivill[128], Khalid Fakhro[129,130,131,132], Hamdi Mbarek[119], Younes Mokrab[129,130,131,133], Jithesh V. Puthen[131], Karsten Suhre[52,56] & Zohreh Tatari[134]**

[124]Microbiome and Biomarkers Discovery Laboratory, Sidra Medicine, Doha, Qatar. [125]College of Health Sciences, Qatar University, Doha, Qatar. [126]Molecular Genetics Laboratory, Hamad Medical Corporation, Doha, Qatar. [127]Department of Genetic Medicine, Microbiology and Immunology, Weill Cornell Medicine–Qatar, Doha, Qatar. [128]Research Branch, Sidra Medicine, Doha, Qatar. [129]Department of Human Genetics, Sidra Medicine, Doha, Qatar. [130]Weill Cornell Medicine–Qatar, Doha, Qatar. [131]College of Health and Life Sciences, Hamad Bin Khalifa University, Doha, Qatar. [132]Genomic Medicine Laboratory, Sidra Medicine, Doha, Qatar. [133]Medical and Population Genomics Laboratory, Sidra Medicine, Doha, Qatar. [134]Clinical Research Centre, Sidra Medicine, Doha, Qatar.

## Methods

### GWAS in each cohort

Following cohort-level quality control of the lung-function pheno-types (Supplementary Note), all phenotypes were rank inverse-normal transformed after adjustment for age, sex, height, smoking, ancestry principal components and relatedness (mixed models in BOLT-LMM or SAIGE). Quality control of the imputation and association summary statistics in each cohort was performed by the central analysis team (Supplementary Note). We assigned each cohort to one of the five 1000 Genomes super-populations—EUR, AFR, AMR, EAS or SAS—based on self-reported ancestry, apart from the UK Biobank (57.4% of the total sample size), where we used ADMIXTURE v1.3.0 (ref. [43]) to determine ancestry (Supplementary Note and Supplementary Table 4). We also acquired lung-function-association results from each cohort using untransformed phenotypes for analysis using MR-MEGA.

### Meta-analysis

Before meta-analysis, association statistics in each cohort were adjusted by the LD-score regression intercept calculated in each cohort to adjust for any residual confounding (Supplementary Table 5); the appropriate ancestry-specific LD reference was used for each cohort (10,000 UK Biobank samples for EUR and 1000 Genomes Project samples for AFR, AMR, SAS and EAS). Before meta-analysis, variants with imputation INFO < 0.5 or minor-allele counts (MAC) < 3 were excluded. As transformed effects were not on comparable scales, we meta-analyzed across cohorts using sample-size weighted $Z$-score meta-analysis with METAL (released version 28 August 2018)[44]. No genomic control was applied post meta-analysis. Following meta-analysis, variants with MAC < 20 were excluded.

### Signal selection and conditional analysis

We chose a genome-wide significance threshold of $P < 5 \times 10^{-9}$, as recommended from sequencing studies[13]. We selected 2-Mb regions centered on the most significant variant for all regions containing a variant with $P < 5 \times 10^{-9}$. Regions within 500 kb of each other were merged for conditional analysis. Stepwise conditional analysis was run in each region in each cohort using GCTA v1.93.2beta[45] with an ancestry-specific LD reference for each cohort (Supplementary Note), and then the conditional results were meta-analyzed across cohorts and any new conditionally independent signals with $P < 5 \times 10^{-9}$ were added to our list of signals. We used moloc v0.1.0 (ref. [46]) to co-localize signals across the four lung-function traits to obtain a set of distinct signals, which were then co-localized with previously reported signals to obtain a set of novel lung-function signals (Supplementary Note).

### Exclusion of smoking signals from follow-up

We checked our sentinels for association with the smoking quantitative traits 'age of initiation' ($n = 262,990$) and 'cigarettes per day' ($n = 263,954$), and the binary traits 'smoking cessation' ($n = 139,453$ cases and $n = 407,766$ controls) and 'smoking initiation' ($n = 557,337$ cases and $n = 674,754$ controls) in the GWAS and Sequencing Consortium of Alcohol and Nicotine use (GSCAN) consortium[47] (proxies with a squared correlation coefficient ($r^2$) > 0.8 were checked for sentinels not present in GSCAN). We excluded eight lung-function signals from further analysis, which we determined to be primarily driven by smoking behavior (Supplementary Table 26), according to the following criteria: (1) $P < 4.86 \times 10^{-5}$ (Bonferroni-corrected 5% threshold for 1,028 signals) for association with any smoking trait and (2) the same 'risk' allele that increases smoking exposure behavior and decreases lung function.

### Heritability estimate

We calculated the proportion of variance explained by the sentinels reported for each trait using the formula

$$\frac{\sum_{i=1}^{n} 2f_i(1-f_i)\beta_i^2}{V}$$

where $n$ is the number of variants, $f_i$ and $\beta_i$ are the frequency and effect estimates of the $i$th variant from the UK Biobank European ancestry untransformed results, respectively, and $V$ is the phenotypic variance (always one as our phenotypes were inverse-normal transformed). We assumed a heritability of 40% (refs. [48,49]) to estimate the proportion of additive polygenic variance.

### Ancestry-adjusted trans-ethnic meta-analysis using MR-MEGA

To improve the fine-mapping resolution using LD differences between ancestries and to estimate the heterogeneity of variant associations attributable to ancestry, we undertook multi-ancestry meta-regression using MR-MEGA v0.2 (ref. [7]), which incorporates axes of genetic ancestry as covariates. MR-MEGA uses multidimensional scaling of allele frequencies across cohorts to derive principal axes of genetic variation to use for ancestry adjustment (Supplementary Note). The location of the cohorts on the first two multidimensional scaling-derived principal components, plotted in Supplementary Fig. 17, shows clustering in accordance with the assigned ancestry groups. We used four principal components for ancestry adjustment, as this captured most of the variance. MR-MEGA implements genomic control at study level; therefore, no further genomic control was applied. We ran MR-MEGA at each locus containing ≥1 signals; in the loci with multiple signals, we ran MR-MEGA multiple times, each time conditioning on all except one signal at the locus. For each sentinel, we obtained an estimated ancestry-associated ($P$-value_ancestry_het) and residual ($P$-value_residual_het) heterogeneity. In addition, MR-MEGA reports the log-transformed Bayes factor, which can be used for the construction of credible sets.

### Effects in children

To obtain unbiased effect estimates for comparison between adults and children, we first redefined 1,077 lead SNPs for lung function in the UK Biobank EUR population ($n = 320,656$) by selecting 1-Mb regions centered on the most significant variant for regions containing a variant with $P < 5 \times 10^{-8}$. For these SNPs, we then took the untransformed effect estimates from the meta-analysis of the non-UK Biobank EUR cohorts (34 cohorts for FEV$_1$ and FVC, $n = 128,071$; 33 cohorts for FEV$_1$/FVC, $n = 123,429$; 15 cohorts for PEF, $n = 60,122$). Next, we meta-analyzed two EUR-ancestry children's cohorts—ALSPAC and Raine Study (age, 13–15 yr, $n = 6,070$)—to obtain effect estimates in children at the new lead SNPs. To investigate the age-dependent effects of genetic variants on lung function, we compared the effect sizes estimated in adults and children using a Welch's $t$-test; a Bonferroni significance threshold for 1,077 tests was applied ($P < 4.64 \times 10^{-5}$).

### Cell-type and functional specificity

**Stratified LD-score regression.** We tested for enrichment of regulatory features at variants overlapping four histone marks (H3K27ac, H3K9ac, H3K4me3 and H3K4me1) that are specific to adult lung, fetal lung, and peripheral blood mononuclear primary and smooth-muscle-containing cell lines (colon and stomach) using stratified LD-score regression[12]. We only considered EUR-specific meta-analysis with 39 cohorts for FVC, FEV$_1$ and FEV$_1$/FVC (17 cohorts for PEF). For the analysis of cell-type-specific annotations, we assessed statistical significance at the 0.05 level after Bonferroni correction for 60 hypotheses tested. Given that these annotations are not independent, a Bonferroni correction is conservative. We also report results with FDR < 0.05 using the Benjamini–Hochberg method.

**Regulatory and functional enrichment using GARFIELD.** We tested enrichment of SNPs at functionally annotated regions (DNase I hypersensitivity hotspots, open chromatin peaks, transcription-factor footprints

and formaldehyde-assisted isolation of regulatory elements, histone modifications, chromatin segmentation states, genic annotations and transcription-factor binding sites) using GARFIELD[17]. We used the EUR meta-analysis with 17 cohorts for PEF and 39 cohorts for FVC, FEV$_1$ and FEV$_1$/FVC. We applied GARFIELD to DNase I hypersensitivity hotspot annotation in 424 cell lines and primary cell types from ENCODE and Roadmap Epigenomics and derived enrichment estimates at trait-genotype association $P$-value thresholds of $P < 5 \times 10^{-5}$ and $P < 5 \times 10^{-9}$.

**Enrichment of annotations in respiratory-relevant cell types and tissues.** We curated annotations from assays of respiratory-relevant cells and tissues—that is, (1) single-cell genome ATAC–seq data[50] from 19 cell types (myofibroblast, pericyte, ciliated, T cell, club, capillary endothelial 1 and 2, basal, matrix fibroblast 1 and 2, arterial endothelial, pulmonary neuroendocrine, natural killer cell, macrophage, B cell, erythrocyte, lymphatic endothelial, alveolar type 1 and 2 (downloaded from https://www.lungepigenome.org/)), (2) ATAC–seq data for five human primary lung-cell types implicated in COPD pathobiology[51] (large and small airway epithelial cells, alveolar type 2, pneumocytes and lung fibroblasts (downloaded from http://www.copdconsortium.org/)) and (3) tissue-specific transcription-factor binding sites from DNase-seq footprinting of 589 human transcription factors in lung and bronchus[52]. We tested for cell- and tissue-specific enrichment of these annotations at our lung-function signals using functional GWAS (fGWAS)[14] (Supplementary Note).

**Identification of putative causal genes and variants**

**eQTL and pQTL co-localization.** Three eQTL resources were used for co-localization of lung-function signals with gene expression signals: (1) GTEx V8 (downloaded from https://www.gtexportal.org/, July 2020; tissues: stomach, small-intestine terminal ileum, lung, esophagus muscularis, esophagus gastroesophageal junction, colon transverse, colon sigmoid, artery tibial, artery coronary and artery aorta), (2) eQTLgen[53] blood eQTLs and (3) UBC lung eQTL[54]. Two blood pQTL resources were used to co-localize with associations with protein levels, that is, INTERVAL pQTL[55] and SCALLOP pQTL. The coloc_susie method[56] was used to test eQTL and pQTL co-localization (Supplementary Note).

**Rare variants from exome sequencing.** We checked for rare (MAF < 1%) exonic associations near (±500 kb) our lung-function sentinels using both single-variant and gene-based collapsing tests from (1) 281,104 UK Biobank exomes from the AstraZeneca PheWAS Portal[57] (https://azphewas.com/), (2) loss-of-function and missense variants in 454,787 UK Biobank participants[58] and (3) gene-based tests on whole-exome imputation in 500,000 UK Biobank participants[59]. We used a threshold of $P < 5 \times 10^{-6}$ for both single-variant and gene-based tests (Supplementary Note).

**Nearby Mendelian respiratory-disease genes.** We selected rare Mendelian-disease genes from ORPHANET (https://www.orpha.net/) within ±500 kb of a lung-function sentinel that were associated with respiratory terms matching regular expression—that is, respir, lung, pulm, asthma, COPD, pneum, eosin, immunodef, cili, autoim, leukopenia, neutropenia and Alagille syndrome. We implicated the gene if it had a corresponding respiratory term match in the disease name or if it occurs frequently in human phenotype ontology terms for that disease (Supplementary Note).

**Nearby mouse-knockout orthologs with a respiratory phenotype.** We selected human orthologs of mouse-knockout genes with phenotypes in the 'respiratory' category, as listed in the International Mouse Phenotyping consortium (https://www.mousephenotype.org/), within ±500 kb of a lung-function sentinel (Supplementary Note).

**PoPS.** We calculated a gene-level PoPS[9] based on the assumption that if the associations enriched in genes share functional characteristics with a gene near to a lung-function signal, then that gene is more likely to be causal. The full set of gene features used in the analysis included 57,543 total features—40,546 derived from gene expression data, 8,718 extracted from a protein–protein interaction network and 8,479 based on pathway membership. In this study we prioritized genes for all autosomal lung-function signals within a 500-kb (±250 kb) window of the sentinel and reported the top prioritized genes in the region. For the signals that did not have prioritized genes within the 500-kb window, we looked for prioritized genes using a 1-Mb (±500 kb) window (Supplementary Note).

**Annotation-informed credible sets.** We used the enriched annotations in respiratory-relevant cell types and tissues and enriched genic annotations (Supplementary Table 12) to create annotation-informed 95% credible sets using fGWAS based on the MR-MEGA ancestry-adjusted meta-regression results (Supplementary Note). We implicated a putative causal missense variant if it accounted for >50% of the posterior probability in the credible set and annotated these using Ensembl Variant Effect Predictor[60] to check for a deleterious effect by the SIFT, PolyPhen or CADD metrics.

**Allocation of genes prioritized with ≥3 variant-to-gene to lung-function biology categories.** We allocated prioritized genes with ≥3 criteria to different lung-function roles (epithelial, inflammatory, peripheral lung (including alveolus and endothelial), lung remodeling (including connective tissue), chest-wall movement and lung development) based on literature reviews, including GeneCards (https://www.genecards.org) and PubMed (https://pubmed.ncbi.nlm.nih.gov). Eighteen of the genes were difficult to assign to a specific category on this basis, mainly because they were involved in generic processes such as transcriptional control in a wide variety of cell types; these are not shown in Supplementary Fig. 12 but are included in Supplementary Table 13.

**Interaction with smoking**
Association testing for lung-function traits (FEV$_1$, FVC, FEV$_1$/FVC and PEF) was calculated separately in ever- and never-smoker subgroups and meta-analyzed across EUR-ancestry cohorts. We included untransformed phenotypes with ever- and never-smoking summary statistics ($n = 28$ cohorts) comprising 206,162 ever-smokers and 229,046 never-smokers. A $z$-test was used to compare genetic effect between the untransformed association results for the ever- and never-smokers:

$$z = \frac{\beta_1 - \beta_2}{\sqrt{se_1^2 + se_1^2}}$$

where se is the standard error of the effect $\beta$. We considered a significant interaction any signal with a $P < 4.9 \times 10^{-5}$ (5% Bonferroni-corrected for 1,020 signals tested).

**GRS**
We selected four ancestry groups in the UK Biobank (UKB) as test datasets (SAS was excluded from GRS analyses because UKB SAS was the only cohort in the multi-ancestry analysis for SAS): UKB EUR, UKB AMR, UKB EAS and UKB AFR. All of the other cohorts except UKB SAS and Qatar Biobank were used as discovery datasets.

We repeated the multi-ancestry meta-regression (MR-MEGA), after excluding the four test GWAS, incorporating the same four axes of genetic variation as covariates to account for ancestry. Autosomal signals for each lung-function trait that were reported in the target ancestry population were included in downstream analysis for each ancestry. For ancestry $j$ ($j$ = EUR, AMR, EAS or AFR), we estimated ancestry-specific predicted allelic effects for the $i$th SNP to be used as weights in the multi-ancestry GRS by

$$\hat{b}_{ij} = \alpha_{0i} + \sum_{k=1}^{4} \alpha_{ki} \bar{x}_{kj}$$

where $\bar{x}_{kj}$ is the averaged position of discovery studies with ancestry $j$ on the $k$th axis of genetic variation from multi-ancestry meta-regression, and $\alpha_{0i}$ and $\alpha_{ki}$ denote the intercept and effect of the $k$th axis of genetic variation for the $i$th SNP from the multi-ancestry meta-regression.

We ran each of the ancestry-specific fixed-effect meta-analyses after excluding the test GWAS from the ancestry group using METAL using the inverse-variance weighting method. For comparison, SNPs used as weights in multi-ancestry GRS were selected to build ancestry-specific GRS for each ancestry.

**Testing GRS in independent COPD case–control cohorts.** We tested the association of multi-ancestry GRS with COPD susceptibility in five EUR-ancestry COPD case–control studies: COPDGene (non-Hispanic white), ECLIPSE, GenKOLS, NETT/NAS and SPIROMICS (non-Hispanic EUR) (Supplementary Table 21). We also tested the association in two AFR ancestry COPD case–control studies: COPDGene (African American) and SPIROMICS (African American) (Supplementary Table 21). Associations were tested using logistic regression models, adjusted for age, age squared, sex, height and principal components. In each COPD case–control study, we divided individuals into deciles according to their weighted GRS. For each decile, logistic models were fitted to compare the risk of COPD for members of the test decile with those with the lowest decile (that is, those with the lowest genetic risk). The results were meta-analyzed by ancestry-specific study groups using the fixed-effect model.

**PheWAS**
We used Deep-PheWAS[40], which addresses both phenotype matrix generation and efficient association testing while incorporating the following developments that are not yet available in current platforms and online resources: (1) clinically curated composite phenotypes for selected health conditions that integrate different data types (including primary and secondary care data) to study phenotypes that are not well captured by current classification trees; (2) integration of quantitative phenotypes from primary care data, such as pathology records and clinical measures; (3) clinically curated phenotype selection for traits that are extremely highly correlated and (4) GRSs. The platform includes 2,421 phenotypes in the UK Biobank, with a subset of 2,243 recommended for association testing—some phenotypes that are generated are used solely in the definition of other phenotypes. We removed the four measures of lung function and added seven phenotypes defined in-house (P4002-6) to give 2,246 as our final maximum number of phenotypes for association. Deep-PheWAS then filters these, requiring a minimum case number; we chose to keep the default settings of a 50-case minimum for binary phenotypes and a 100-case minimum for quantitative phenotypes. After limiting to EUR ancestry and filtering for case numbers, 1,909 phenotypes were left for association analysis (Supplementary Table 27). No additional phenotypes were removed when removing pairs related up to second degree (KING kinship coefficient ≥ 0.0884).

There are five types of phenotypes within Deep-PheWAS categorized according to the data and methods used to create them. Composite phenotypes are made using linked hospital and primary care data, including in some cases primary care prescription data, alongside any of the UK Biobank field-IDs (DFP), including self-reported non-cancer diagnosis and self-reported operations. Phecodes are defined using only linked hospital data (https://phewascatalog.org/phecodes_icd10). Formula phenotypes combine available data using bespoke R code per phenotype rather than the in-built functions of phenotype development available in Deep-PheWAS. Added phenotypes are lists of cases and controls that have been added to the PheWAS and not developed by the Deep-PheWAS phenotype matrix generation pipeline. More complete definitions for all none-added phenotypes can be found in the Deep-PheWAS description[40]. All phenotypes were adjusted for age, sex and the first ten principal components.

**Single-variant PheWAS.** We ran 28 single-variant PheWAS across 1,909 traits (Supplementary Table 27) in up to 430,402 unrelated EUR individuals in the UK Biobank. We selected the variant with the most significant $P$ value for each of the 20 genes with ≥4 lines of evidence for being causal (Supplementary Table 13). A further seven variants were included in single-variant PheWAS that were putatively causal (accounted for >50% posterior probability in the credible set and had a deleterious annotation; Supplementary Table 14) but in a gene that was implicated by fewer than four lines of evidence. The single-variant PheWAS was aligned to the lung-function-trait decreasing allele. Where we noted associations with testosterone and SHBG, we also undertook sex-stratified PheWAS.

**Association with trait-specific GRS.** We created four GRSs for the UK Biobank EUR samples, one for each trait FEV$_1$, FVC, FEV$_1$/FVC and PEF, including all conditionally independent sentinel variants for the trait that were associated with $P < 5 \times 10^{-9}$, yielding 425, 372, 442 and 194 variants in each trait-specific GRS, respectively. Each of the four GRS were weighted by the effect sizes from the multi-ancestry meta-regression for the relevant trait and then checked for association with 1,909 traits in the PheWAS.

**Association with pathway-specific GRS.** We selected 29 pathways that were enriched at FDR $< 10^{-5}$ for our 559 genes implicated by ≥2 lines of evidence (Supplementary Table 18). We created a weighted GRS (weights estimated from multi-ancestry meta-regression for FEV$_1$/FVC) for each of the 29 pathways by including for each gene in the pathway (as for 'Single-variant PheWAS') the variant with the most significant $P$ value for the trait that implicates the gene in our variant-to-gene mapping (Supplementary Table 13). Each of the 29 GRSs were then checked for association with 1,909 traits in the PheWAS.

**Reporting summary**
Further information on research design is available in the Nature Portfolio Reporting Summary linked to this article.

## Data availability
Genome-wide summary statistics for the multi-ancestry meta-analysis are available at the GWAS Catalog (https://www.ebi.ac.uk/gwas/) under the accession codes GCST90244092, GCST90244093, GCST90244094 and GCST90244095.

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

## Acknowledgements

This research has been conducted using the UK Biobank Resource under Application Number 648. We used the ALICE High Performance Computing Facility at the University of Leicester. This study was supported by BREATHE–The Health Data Research Hub for Respiratory Health (grant no. MC_PC_19004). This study was partially supported by the NIHR Leicester Biomedical Research Centre and the NIHR Nottingham Biomedical Research Centre; views expressed are those of the author(s) and not necessarily those of the NHS, the NIHR or the Department of Health. The funders had no role in the design of the study. This research was funded in part by the Wellcome Trust. For the purpose of open access, the authors have applied a CC BY public copyright license to any author accepted paper version arising from this submission. Cohort- and study-group-specific acknowledgements and funding information are in the Supplementary Note. This study was also supported by specific personal funding from the following: Wellcome Trust Institutional Strategic Support Fund (grant no. 204801/Z/16/Z) and BHF Accelerator Award AA/18/3/34220 to A.L.G.; NIH K08 HL136928, U01 HL089856, R01 HL155749 and a Research Grant from the Alpha-1 Foundation to B.D.H., B.D.H. also receives grant support from Bayer; Medical Research Council Clinical Research Training Fellowship (grant no. MR/P00167X/1) to C.J.; National Health and Medical Research Council (NHMRC) Ideas grant no. 2003629 and Department of Health Western Australia Merit Award 1186046 to B.H.M.; BBSRC CASE studentship with GSK to A.T.W.; GSK/British Lung Foundation Chair in Respiratory Research (L.V.W.); Wellcome Trust Investigator Award (WT202849/Z/16/Z) and Wellcome Trust Discovery Award (WT225221/Z/22/Z) to M.D.T.; MRC grant no. MR/N011317/1 to M.D.T. and L.V.W.; NIHR Senior Investigator Award to M.D.T. and I.P.H.; MRC grant no. G1000861 to I.S.; Wellcome Trust grant nos. WT098017 and WT064890 to A.P.M.; UKRI Innovation Fellowship at Health Data Research UK grant no. MR/S003762/1 to C.B.; MRC Human Genetics Unit program grant 'Quantitative traits in health and disease' (grant no. U. MC_UU_00007/10) to C.H., V.V. and S.M.K.; Academy of Finland (grant no. 336823) and Sigrid Juselius Foundation to J.K.; and MRC grant no. MR/P009581/1 to E.L.R.N.J.T. is a Wellcome Trust Investigator (202802/Z/16/Z), is the PI of the Avon Longitudinal Study of Parents and Children (MRC & WT 217065/Z/19/Z), is supported by the University of Bristol NIHR Biomedical Research Centre (BRC-1215-2001) and the MRC Integrative Epidemiology Unit (MC_UU_00011/1) and works within the CRUK Integrative Cancer Epidemiology Programme (C18281/A29019).

## Author contributions

M.D.T. and I.P.H. supervised the study. M.D.T., I.P.H., N.S. and A.L.G. designed the study. N.S., A.G.I., J.C., R.P., R.J.H., R.J.T., C. Batini, K.A.F., C.J., K.C., N.N.P. and A.T.W. did the central analysis. F.D., A.P.M. and L.V.W. provided methodological and statistical advice. J.-H.L, I.S. and E.L.R. provided functional analyses. M.D.T., N.S., A.G.I., J.C., R.P. and I.P.H. wrote the paper. The following authors supervised or ran the analysis in the cohorts listed (full cohort names in Supplementary Note): ALHS cohort, A.B.W., J.J., S.S., M.L. and S.J.L.; ALSPAC cohort, R.G. and N.J.T.; ARIC cohort, M.R.B. and A.C.M.; B58C cohort, D.P.S.; BHS cohort, B.H.M., J.H., A. James and J.B.; Boston cohort, C.P., V.M., B.D.H., M.M., W.K., R.T.-S., P.B., E.K.S. and M.H.C.; CHS cohort, T.M.B. and S.A.G.; CKB cohort, A. Pozarickij, K.L., I.Y.M., Zhengming Chen, L. Li and R.G.W.; Croatia cohort, S.M.K., V.V., O.P. and I.R.; EPIC cohort, J. Luan, J.-H.Z. and C.L.; EXCEED cohort, A.T.W., R.C.F., X.W. and C. Brightling; FHS cohort, E.L., H.X., G.T.O. and J.D.; FinnTwin cohort, T.Palviainen, J.K., T.R. and K.H.P.; GS cohort, D.P., B.H.S., A.C. and C.H.; H2000 cohort, J.T.K., T.V., T. Laitinen and V.S.; HCHS cohort, T.S., J. Lee, M.L.D. and B.Y.; HUNT cohort, A.F.H., A.L., C.W., L.B. and B.B.; KORA cohort, C.F., S.K., C.G. and A. Peters; LBC1936 cohort, S.E.H., A.T., P.R. and S.R.C.; MESA cohort, A. Manichaikul, E.C.O., S.S.R. and R.G.B.; NEO cohort, T.O.F., P.S.H., A.M.S. and D.O.M.-K.; NFBC cohort, M.W. and M.-R.J.; ORCADES cohort, S.M.-W., N.P. and J.F.W.; PIVUS cohort, J.P.C., A. Mahajan, L. Lind and A.P.M.; QBB cohort, G.T., O.M.E.A. and K.S.; Raine Study, C.A.W., R.E.F., C.E.P. and G.L.H.; Rotterdam cohort, S.R.A.W., L. Lahousse, G.B. and A.G.U.; SAPALDIA cohort, M.I., A. Jeong and N.P.-H.; SHIP cohort, B.S., S.W., G.H. and C.O.S.; SPIROMICS cohort, X.H., V.E.O., D.A.M., E.R.B., S.B.G., N.G. and A.V.S.; TwinsUK cohort, K.P. and P.H.; UKHLS cohort, E.Z.; USC cohort, J.L.M., F.D.G., Zhanghua Chen and W.J.G.; VIKING cohort, K.A.K., P.K.J. and P.R.H.J.T.; and YFS cohort, T. Lehtimäki and O.T.R. All co-authors critically reviewed the paper.

## Competing interests

M.D.T. and L.V.W. have previously received funding from GSK for collaborative research projects outside of the submitted work. I.P.H. has funded research collaborations with GSK, Boehringer Ingelheim and Orion. M.H.C. has received grant funding from GSK and Bayer, and speaking or consulting fees from AstraZeneca, Illumina and Genentech. B.D.H. has received grant funding from Bayer and speaking or consulting fees from AstraZeneca. I.S. has funded research collaborations with GSK, Boehringer Ingelheim and Orion outside of the submitted work. R.J.P., M.D.T., C.J. and L.V.W. have a funded research collaboration with Orion for collaborative research projects outside of the submitted work. The other authors declare no competing interests.

## Additional information

**Correspondence and requests for materials** should be addressed to Nick Shrine or Martin D. Tobin.

# Reporting Summary

## Statistics

For all statistical analyses, confirm that the following items are present in the figure legend, table legend, main text, or Methods section.

| n/a | Confirmed | |
|---|---|---|
| ☐ | ☒ | The exact sample size (*n*) for each experimental group/condition, given as a discrete number and unit of measurement |
| ☐ | ☒ | A statement on whether measurements were taken from distinct samples or whether the same sample was measured repeatedly |
| ☐ | ☒ | The statistical test(s) used AND whether they are one- or two-sided<br>*Only common tests should be described solely by name; describe more complex techniques in the Methods section.* |
| ☐ | ☒ | A description of all covariates tested |
| ☐ | ☒ | A description of any assumptions or corrections, such as tests of normality and adjustment for multiple comparisons |
| ☐ | ☒ | A full description of the statistical parameters including central tendency (e.g. means) or other basic estimates (e.g. regression coefficient) AND variation (e.g. standard deviation) or associated estimates of uncertainty (e.g. confidence intervals) |
| ☐ | ☒ | For null hypothesis testing, the test statistic (e.g. $F$, $t$, $r$) with confidence intervals, effect sizes, degrees of freedom and $P$ value noted<br>*Give P values as exact values whenever suitable.* |
| ☐ | ☒ | For Bayesian analysis, information on the choice of priors and Markov chain Monte Carlo settings |
| ☒ | ☐ | For hierarchical and complex designs, identification of the appropriate level for tests and full reporting of outcomes |
| ☐ | ☒ | Estimates of effect sizes (e.g. Cohen's *d*, Pearson's *r*), indicating how they were calculated |

*Our web collection on statistics for biologists contains articles on many of the points above.*

## Software and code

Policy information about availability of computer code

| Data collection | None |
|---|---|
| Data analysis | BOLT-LMM v2.3.4, R v4.1.0, ADMIXTURE v1.3.0, METAL v2018-08-28, LDSC v1.0.0, GCTA v1.93.2beta, PLINK v1.9, PLINK v2.0, ANNOVAR v2014-11-12, MR-MEGA v0.2, GARFIELD v2, fGWAS v0.3.6, coloc v3.2-1 |

For manuscripts utilizing custom algorithms or software that are central to the research but not yet described in published literature, software must be made available to editors and reviewers. We strongly encourage code deposition in a community repository (e.g. GitHub). See the Nature Portfolio guidelines for submitting code & software for further information.

## Data

Policy information about availability of data

All manuscripts must include a data availability statement. This statement should provide the following information, where applicable:
- Accession codes, unique identifiers, or web links for publicly available datasets
- A description of any restrictions on data availability
- For clinical datasets or third party data, please ensure that the statement adheres to our policy

Genome-wide summary statistics for the multi-ancestry meta-analysis are available at GWAS catalog  (https://www.ebi.ac.uk/gwas/); accession codes GCST90244092, GCST90244093, GCST90244094, GCST90244095

# Human research participants

Policy information about studies involving human research participants and Sex and Gender in Research.

| | |
|---|---|
| Reporting on sex and gender | Self-reported sex was used as a covariate in the GWAS in each of the 49 contributing cohorts. In total there were 256,850 males and 324,055 females included in this study. Individual level sex information is not available. |
| Population characteristics | Genotypic information, spirometry, age, sex, height and ancestry principal components were used in the 49 contributing cohorts and questionnaire results and primary and secondary care diagnoses were used for the PheWAS in UK Biobank. Covariates used in association testing were age (at spirometry measurement), age2, sex, and height, and also axes of genetic ancestry to model heterogeneity due to ancestry. Covariate-relevant population characteristics are summarised in Supplementary Table 2. |
| Recruitment | Recruitment of participants in the 49 cohorts is described in the Supplementary Note. |
| Ethics oversight | The organisations approving the study protocol in each of the cohorts is given in the Supplementary Note. |

Note that full information on the approval of the study protocol must also be provided in the manuscript.

# Field-specific reporting

Please select the one below that is the best fit for your research. If you are not sure, read the appropriate sections before making your selection.

☒ Life sciences   ☐ Behavioural & social sciences   ☐ Ecological, evolutionary & environmental sciences

For a reference copy of the document with all sections, see nature.com/documents/nr-reporting-summary-flat.pdf

# Life sciences study design

All studies must disclose on these points even when the disclosure is negative.

| | |
|---|---|
| Sample size | In order to maximise power, we used all samples available in 49 contributing cohorts with good measures of lung function and that had been genotyped giving a total sample size of 580,869 |
| Data exclusions | Cohorts excluded samples with low-quality lung function measurements or samples that failed genotyping quality control. |
| Replication | To maximise power, all cohorts were used in discovery and we assessed the heterogeneity of effect estimates across contributing cohorts, and the extent to which heterogeneity was attributable to ancestry. After accounting for heterogeneity due to ancestry 93 of 1020 signals reported showed residual heterogeneity of effects across cohorts. |
| Randomization | This was an observational study; due to random allocation of genetic variants during gamete production genetic association studies of germline association are not expected to be subject to the confounding and reverse causation typically seen in traditional observational epidemiology studies |
| Blinding | Blinding was not necessary as the analysts were not involved in the measurement of lung function or genotypes and all samples are anonymised. |

# Reporting for specific materials, systems and methods

We require information from authors about some types of materials, experimental systems and methods used in many studies. Here, indicate whether each material, system or method listed is relevant to your study. If you are not sure if a list item applies to your research, read the appropriate section before selecting a response.

## Materials & experimental systems

| n/a | Involved in the study |
|---|---|
| ☒ ☐ | Antibodies |
| ☒ ☐ | Eukaryotic cell lines |
| ☒ ☐ | Palaeontology and archaeology |
| ☒ ☐ | Animals and other organisms |
| ☒ ☐ | Clinical data |
| ☒ ☐ | Dual use research of concern |

## Methods

| n/a | Involved in the study |
|---|---|
| ☒ ☐ | ChIP-seq |
| ☒ ☐ | Flow cytometry |
| ☒ ☐ | MRI-based neuroimaging |

