## [Peer Review File · Nature Genetics]

Peer Review Information

Manuscript Title: Multi-ancestry genome-wide association analyses improve resolution of genes and pathways influencing lung function and chronic obstructive pulmonary disease risk

Corresponding author name(s): Dr Nick Shrine, Professor Martin Tobin

Reviewer Comments & Decisions:

Decision Letter, initial version:
--

9th August 2022

Dear Nick,

Your Article "Multi-ancestry genome-wide association study improves resolution of genes, pathways and pleiotropy for lung function and chronic obstructive pulmonary disease" has been seen by two referees. You will see from their comments below that, while they find your work of interest, they have raised several relevant points. We are interested in the possibility of publishing your study in Nature Genetics, but we would like to consider your response to these points in the form of a revised manuscript before we make a final decision on publication.

To guide the scope of the revisions, the editors discuss the referee reports in detail within the team with a view to identifying key priorities that should be addressed in revision, and sometimes overruling referee requests that are deemed beyond the scope of the current study. In this case, we ask that you carefully revise the presentation and interpretation of the results throughout to address the specific concerns highlighted by the referees, including modifying the figures for clarity and providing appropriate statistical analyses where needed. We hope you will find this prioritized set of referee points to be useful when revising your study. Please do not hesitate to get in touch if you would like to discuss these issues further.

We therefore invite you to revise your manuscript taking into account all reviewer and editor comments. Please highlight all changes in the manuscript text file. At this stage, we will need you to upload a copy of the manuscript in MS Word .docx or similar editable format.

*2) If you have not done so already please begin to revise your manuscript so that it conforms to our Article format instructions, available [here](http://www.nature.com/ng/authors/article_types/index.html). Refer also to any guidelines provided in this letter.

[redacted]

We hope to receive your revised manuscript within 4-8 weeks. If you cannot send it within this time, please let us know.

Sincerely,
Kyle

Kyle Vogan, PhD
Senior Editor
Nature Genetics
<https://orcid.org/0000-0001-9565-9665>

Referee expertise:

Referee #1: Genetics, pulmonary diseases

Referee #2: Genetics, pulmonary diseases, statistical methods

Reviewers' Comments:

Reviewer #1:
Remarks to the Author:

The authors performed a multi-ancestry GWAS meta-analysis of lung function comprising 580,869 participants. The authors found 1,020 independent association signals and identified 559 candidate genes, which were enriched in 29 pathways. The authors also conducted PheWAS for selected associated variants, and trait and pathway-specific GRS to infer possible consequences of intervening in pathways underlying lung function. Individual variants collectively as a GRS showed a strong association with COPD across ancestry groups. The authors identified new putative causal variants, proteins, and pathways including potential drug targets for the treatment of COPD. I have the following comments.

Major comments

1) The authors performed GWAS of lung functions from 49 cohorts and found a total of 713 novel signals with respect to the signals and studies. It is difficult to find these reported GWAS studies in the Supplementary Note. It would be better to briefly summarize the studies with references in a supplementary table.

It would be informative to indicate the number of novel signals with common or specific in European and non-European populations. Throughout, new findings in non-European populations should be highlighted. A recent study has reported a cross-population atlas of genetic associations for 220 human phenotypes (Nat Genet. 2021 Oct;53(10):1415-1424). The summary statistics of COPD GWAS in the Japanese population (4,017 cases and 162,653 controls, non-European population) are available on the PheWeb.jp. I thought it would be better to discuss the relationship between the novel findings in this study and the COPD GWAS.

Descriptions of spirometric quality control should be added to the main text.

2) In this study, the authors conducted an LD score regression analysis of four lung function traits and found heritability enrichment at lung and smooth-muscle-specific histone marks. It would be better to examine blood cells or epithelial cells.

3) In Figure 2, I think it is easier to understand if the genes are listed in the table from the one with the largest number to the one with the fewest "variant-to-gene criteria" (5 to 3). It seems that the information in Figure 3 can be integrated into Figure 2. Locus plots for representative loci containing candidate genes that meet 5 or 4 variant-to-gene criteria would be informative.

4) In Figure 5d, the authors have described that the odds ratio for COPD showed steeper increase as the GRS decile increases using the multi-ancestry GRS compared with the previous GRS (2019). Statistical evidence would seem to be needed.

5) Since the results of PheWAS of lung function trait GRS in Figure 6 are difficult to see and understand, it would be better to show the list of phenotypes in the top 10 including the direction of effect in a larger size in the upper right part of the figure.

In Figure 7, to indicate the list of top 10 phenotypes in the figure would also be helpful. In Supplementary Figure 13c, the authors just show pairwise scatter plots of individual pathway genetic risk scores for four pathways. Further descriptions are needed for the results including statistical analysis.

Minor comments

1) In Supplementary Figure 3, it would be better to indicate sentinels in the novel loci than show the reported sentinels. Descriptions of the red line and the blue line should be added.

2) In Supplementary Figure 4, it is difficult to read black characters within dark-colored markers.

3) Figure 4 was cited in the main text prior to Figures 2 and 3. Supplementary tables 4,5, 8, and 13 should be cited in order in the main text. Please check the order of the Figures, supplementary figures, and tables carefully.

4) Supplementary Figure 8b, the upper part of the figure has been cut off. Some of the text in the figure is too small to read.

5) There are several places where text and text or lines overlap in Figure 6 and Supplementary Figure 14.

6) References of Multi-ancestry paper Supplement weren't written in the unified style. Please check carefully.

Reviewer #2:

Remarks to the Author:

The paper describes an important meta-analysis of lung function phenotypes that sheds new light into the genetics of these traits and COPD. I think the paper would benefit from a rewriting that keeps only the most relevant results. Some comments on issues that I found are below.

Specific comments:

There is a statement in the second paragraph of results that I am not convinced is true: "These 1020 signals show a pattern of increasing effect size as allele frequency decreases, in keeping with other complex traits". The sentence seems to imply that rare alleles tend to have higher effects (implying some sort of evolutionary selection on a trait that is related to lung function). I believe that this is just an outcome of "winner's curse": for the rare SNPs to make the significance cutoff, their sample effect size is a huge overestimation of the true effect size. Moreover, effect size are clearly different in different ancestry groups and what the authors call "effect size" is a weighted average of ancestry-specific effect sizes. This sentence needs to be rephrased or an accurate analysis of effect size needs to be performed.

A similar misinterpretation is in the "Effects in children" analysis. The authors compare effect sizes in children (calculated, it seems, on data from studies that were not used for GWAS) to effect sizes in adults (calculated from studies used in GWAS, and hence overestimated). The comparison is flawed because the adult data used in this comparison was part of the discovery analysis (if my understanding of which data was used is correct). You can see this clearly in Supplementary Figure 9 where the adult effect sizes tend to be larger than those in children (see above paragraph). Moreover, the correlation shown is also wrong (for the same reason) - in fact it might be underestimated. Note that this issue affects other analyses, for example what is shown in Supplementary Figure 10 (the issue there is that correlation is the improper statistic).

There are some exploratory analyses for which it is difficult to evaluate their significance. For example, in the results section "Identification of putative causal genes and causal variants", SNPs were cross-checked with other databases (eQTLs, etc.). The results seem to be over-stated because there is no proper statistical analysis of enrichment. For example, I don't know what is the baseline for Figure 2. If one would do the same analysis for a set of random SNPs (matched on imputation scores, MAF, etc.), what proportion of them would satisfy the enrichment criteria used there (3 variant-to-gene, for example)? Without an understanding of that, what is meaning of these results?

I am not sure what are the conclusions from the "Phenome-wide association" analysis (Figure 6). What is the most interesting discovery there and what is its meaning? Is the fact that height is associated with these GRSs a signal that the phenotypes were not properly normalized? Moreover, "Pack years" show up in the FVC analysis...

The "Druggable targets" section seems very speculative - the evidence for repurposing vatelizumab is weak.

Data sharing: are the data or summary statistics shared? I could not find data sharing section, but it is possible I missed it. I am sure NG has data sharing and reproducibility policies.

There is an excess of Figures and not all of them are carefully chosen:

- Figure 3 can be summarized in a small table. I am not sure if all the nice drawings are needed and the rectangles pointing to the lungs seems to imply some connections (e.g. inflammation and lung remodeling) that might not be relevant. Also, why have two full-page figures (Figures 2 and 3) for such a small part of the paper when they contain a lot of overlapping information? I recommend choosing one.

- Figure 4 is difficult to read because of font issues (names of genes and rs numbers cannot be read on a laptop-sized screen). I think the information would be more useful in supplementary table.

- Figure 6: there must be a better way to visualize these results. I learned everything on these analyses from the text description and I did not get too much from figures. Text is not readable for some of them (e.g. Figure 6c). Minor: why is asthma in a group called "Safety"? Similar comments for Figure 7.

Minor: maybe nobody reads supplementary materials anymore, but reading the ones for this paper requires major effort. Are all the figures necessary, or is the supplement a dumping ground for all analyses that were done (even if they yielded nothing interesting)?

Author Rebuttal to Initial comments

Response to Reviewers

Reviewer #1:

Remarks to the Author:

The authors performed a multi-ancestry GWAS meta-analysis of lung function comprising 580,869 participants. The authors found 1,020 independent association signals and identified 559 candidate genes, which were enriched in 29 pathways. The authors also conducted PheWAS for selected associated variants, and trait and pathway-specific GRS to infer possible consequences of intervening in pathways underlying lung function. Individual variants collectively as a GRS showed a strong association with COPD across ancestry groups. The authors identified new putative causal variants, proteins, and pathways including potential drug targets for the treatment of COPD. I have the following comments.

Major comments

1) The authors performed GWAS of lung functions from 49 cohorts and found a total of 713 novel signals with respect to the signals and studies. It is difficult to find these reported GWAS studies in the Supplementary Note. It would be better to briefly summarize the studies with references in a supplementary table.

Thank you for highlighting this. We have added a column to Supplementary Table 6 to show the previously reported variant and the PubMed ID or DOI for the study that reported each previously reported finding.

It would be informative to indicate the number of novel signals with common or specific in European and non-European populations. Throughout, new findings in non-European populations should be

highlighted. A recent study has reported a cross-population atlas of genetic associations for 220 human phenotypes (Nat Genet. 2021 Oct;53(10):1415-1424). The summary statistics of COPD GWAS in the Japanese population (4,017 cases and 162,653 controls, non-European population) are available on the PheWeb.jp. I thought it would be better to discuss the relationship between the novel findings in this study and the COPD GWAS.

In assessing novelty of our genetic signals, we curated signals from previously published studies passing a threshold of $P < 5 \times 10^{-9}$, now additionally including 4 signals from the COPD GWAS in the Japanese population (Supplementary Note page 21). None of these 4 signals overlapped with the genome-wide significant signals we report in our multi-ancestry analysis. We have made edits to the discussion to acknowledge the limitation of our focus on multi-ancestry discovery rather than ancestry-specific signals, and to discuss the steps to be taken to improve power for ancestry-specific signal discovery in the future.

Descriptions of spirometric quality control should be added to the main text.

We now refer to cohort-level quality control in the Methods section, cross-referencing the Supplementary Note's description of each cohort and we have added text on the spirometry QC in UK Biobank to page 8 of the Supplementary note.

2) In this study, the authors conducted an LD score regression analysis of four lung function traits and found heritability enrichment at lung and smooth-muscle-specific histone marks. It would be better to examine blood cells or epithelial cells.

We tested for enrichment of regulatory features overlapping 4 different histone marks in 4 cell types (lung, fetal lung, colon smooth muscle and stomach smooth muscle) for 4 lung function traits – this is now made clearer to which we have now added primary mononuclear cells. No significant enrichment was shown in mononuclear primary cell type and the conclusion of heritability enrichment at other cell type-specific histone marks stayed the same at Bonferroni correction (see updated Supplementary Table 17). A few mononuclear primary-specific histone marks showed $FDR < 0.05$ (using Benjamini-Hochberg method) for some lung function traits. We were unable to identify relevant epithelial cells with such annotations (https://alkesgroup.broadinstitute.org/LDSCORE/1000G_Phase1_cell_type_ldscores.tgz).

3) In Figure 2, I think it is easier to understand if the genes are listed in the table from the one with the largest number to the one with the fewest “variant-to-gene criteria” (5 to 3). It seems that the information in Figure 3 can be integrated into Figure 2. Locus plots for representative loci containing candidate genes that meet 5 or 4 variant-to-gene criteria would be informative.

The genes in Figure 2 were already ordered as the reviewer requested (5 to 3) but we have made this clearer by adding the number of variant-to-gene criteria in brackets after the gene name. We have

moved Figure 3 to the supplement (Supplementary Figure 16) in view of the recommendations of both reviewers.

4) In Figure 5d, the authors have described that the odds ratio for COPD showed steeper increase as the GRS decile increases using the multi-ancestry GRS compared with the previous GRS (2019). Statistical evidence would seem to be needed.

Thank you for highlighting this. We have reworded the text to clarify the hypothesis being tested (bottom of page 9) and have provided the statistical test results in Figure 5d.

5) Since the results of PheWAS of lung function trait GRS in Figure 6 are difficult to see and understand, it would be better to show the list of phenotypes in the top 10 including the direction of effect in a larger size in the upper right part of the figure. In Figure 7, to indicate the list of top 10 phenotypes in the figure would also be helpful.

In the PheWAS figures (now Figures 4 & 5), we have reduced the number of labels and improved the size of labels to improve clarity.

In Supplementary Figure 13c, the authors just show pairwise scatter plots of individual pathway genetic risk scores for four pathways. Further descriptions are needed for the results including statistical analysis.

We apologise that Supplementary Figure 13 and its legend were not as clear as they should have been. We have improved the legend and the pathway labels, and now show the correlation coefficient and its p-value for pairwise comparisons of pathway-specific GRS.

Minor comments

1) In Supplementary Figure 3, it would be better to indicate sentinels in the novel loci than show the reported sentinels. Descriptions of the red line and the blue line should be added.

Thank you. the revised Supplementary Figure 3 now highlights the novel signals, and its legend has been amended accordingly.

2) In Supplementary Figure 4, it is difficult to read black characters within dark-colored markers.

We agree. Supplementary Figure 4 has been given clearer contrast and improved size of labels.

3) Figure 4 was cited in the main text prior to Figures 2 and 3. Supplementary tables 4,5, 8, and 13 should be cited in order in the main text. Please check the order of the Figures, supplementary figures, and tables carefully.

We apologise for this oversight. The citations and ordering have been corrected, accounting for the content moved to the Supplement.

4) *Supplementary Figure 8b, the upper part of the figure has been cut off. Some of the text in the figure is too small to read.*

Thank you for highlighting this. It has been corrected.

5) *There are several places where text and text or lines overlap in Figure 6 and Supplementary Figure 14.*

We have improved the clarity of these figures as suggested.

6) *References of Multi-ancestry paper Supplement weren't written in the unified style. Please check carefully.*

Thank you. These have been corrected.

Reviewer #2:

Remarks to the Author:

The paper describes an important meta-analysis of lung function phenotypes that sheds new light into the genetics of these traits and COPD. I think the paper would benefit from a rewriting that keeps only the most relevant results. Some comments on issues that I found are below.

Specific comments:

There is a statement in the second paragraph of results that I am not convinced is true: "These 1020 signals show a pattern of increasing effect size as allele frequency decreases, in keeping with other complex traits". The sentence seems to imply that rare alleles tend to have higher effects (implying some sort of evolutionary selection on a trait that is related to lung function). I believe that this is just an outcome of "winner's curse": for the rare SNPs to make the significance cutoff, their sample effect size is a huge overestimation of the true effect size. Moreover, effect size are clearly different in different ancestry groups and what the authors call "effect size" is a weighted average of ancestry-specific effect sizes. This sentence needs to be rephrased or an accurate analysis of effect size needs to be performed. A similar misinterpretation is in the "Effects in children" analysis. The authors compare effect sizes in children (calculated, it seems, on data from studies that were not used for GWAS) to effect sizes in adults (calculated from studies used in GWAS, and hence overestimated). The comparison is flawed because the adult data used in this comparison was part of the discovery analysis (if my understanding of which data was used is correct). You can see this clearly in Supplementary Figure 9 where the adult effect sizes tend to be larger than those in children (see above paragraph). Moreover, the correlation shown is also wrong (for the same reason) - in fact it might be underestimated. Note that this issue affects other analyses, for example what is shown in Supplementary Figure 10 (the issue there is that correlation is the improper statistic).

A pattern of increasing effect size as allele frequency decreases has been widely observed and previously reported across many complex traits, including the study we reference by Park et al 2011 (PMID: 22003128), which adopted approaches to exclude winners curse bias as an explanation.

Subsequent papers confirm this pattern across complex traits and suggest (as the reviewer highlights) that low allele frequencies tend to result from negative selection of variants of large effects (Schoech et al PMID: 30770844). Whilst re-examination of negative selection in complex traits is beyond the scope of our paper, we think it is appropriate to refer to this earlier work and to the consistency of our findings with it. We have added the reference to Schoech et al PMID: 30770844.

We do not concur that correlation coefficients of effect size estimates (between children and adults, for example) are wrong to present, although we agree with the reviewer that care is needed in the interpretation of the findings. We acknowledge that we cannot rule out winner's curse bias affecting the comparisons highlighted and we have added a statement to the Discussion: "The use of available datasets for discovery rather than follow-up means that we cannot rule out possible inflation of effect size estimates due to winner's curse bias, which in turn may affect effect size comparisons, such as those between adults and children or between ancestry groups." (bottom of p11, top of p12)

There are some exploratory analyses for which it is difficult to evaluate their significance. For example, in the results section "Identification of putative causal genes and causal variants", SNPs were cross-checked with other databases (eQTLs, etc.). The results seem to be over-stated because there is no proper statistical analysis of enrichment. For example, I don't know what is the baseline for Figure 2. If one would do the same analysis for a set of random SNPs (matched on imputation scores, MAF, etc.), what proportion of them would satisfy the enrichment criteria used there (3 variant-to-gene, for example)? Without an understanding of that, what is meaning of these results?

The purpose of the variant-to-gene mapping approaches is not to describe enrichment, but to provide biological interpretation for individual signals and prioritise genes and variants for functional experiments. The individual variant-to-gene mapping approaches we describe such as eQTL, pQTL, credible sets, PoPs, WES associations all have rigorous statistical analyses and criteria which are described in the paper. We accept that approaches to aggregate the different types of variant-to-gene mapping evidence are less well-developed, and we have added the following to the discussion: "Such frameworks, whilst guiding prioritisation of genes for functional experiments, do not provide definitive guidance on how variant-to-gene criteria should be weighted or on a minimal number of variant-to-gene criteria required. Such in-silico evidence cannot firmly demonstrate causality, and confirmation of mechanism will require functional genomics experiments such as gene editing in suitable organoids with appropriate readouts."

I am not sure what are the conclusions from the "Phenome-wide association" analysis (Figure 6). What is the most interesting discovery there and what is its meaning? Is the fact that height is associated with these GRSs a signal that the phenotypes were not properly normalized? Moreover, "Pack years" show up in the FVC analysis...

Please see comments below regarding Figure 6 and its interpretation.

The "Druggable targets" section seems very speculative - the evidence for repurposing vatelizumab is weak.

Genetic evidence of drug target effects on relevant phenotypes improves, but does not guarantee, success of drug repurposing or new drug efficacy. We have therefore employed cautious phrasing when interpreting such evidence. In the results, the evidence is presented alongside the statement "indicating a *potential* to repurpose vatelizumab", and we have removed the mention of vatelizumab from the Discussion.

Data sharing: are the data or summary statistics shared? I could not find data sharing section, but it is possible I missed it. I am sure NG has data sharing and reproducibility policies.

The full GWAS summary statistics will be available via the GWAS Catalog after acceptance of the paper (as for our previous paper, <https://www.ebi.ac.uk/gwas/publications/30804560>). We have added a statement to the supplement to this effect (top of p.8 in Supplement).

There is an excess of Figures and not all of them are carefully chosen:

- Figure 3 can be summarized in a small table. I am not sure if all the nice drawings are needed and the rectangles pointing to the lungs seems to imply some connections (e.g. inflammation and lung remodeling) that might not be relevant. Also, why have two full-page figures (Figures 2 and 3) for such a small part of the paper when they contain a lot of overlapping information? I recommend choosing one.

Thank you. We have revised Figure 2 and moved Figure 3 to the supplement (now Supplementary Figure 16) in view of the recommendations of both reviewers.

- Figure 4 is difficult to read because of font issues (names of genes and rs numbers cannot be read on a laptop-sized screen). I think the information would be more useful in supplementary table.

Thank you for this suggestion. We have removed the figure from the main paper. We provide relevant information in Supplementary Table 8 and as the figure contains information difficult to convey in a single table, we have retained this as Supplementary Figure 17.

- Figure 6: there must be a better way to visualize these results. I learned everything on these analyses from the text description and I did not get too much from figures. Text is not readable for some of them (e.g. Figure 6c). Minor: why is asthma in a group called "Safety"? Similar comments for Figure 7.

We are pleased that the text interpreting these figures and summarising the salient findings was informative. We have improved the clarity of the PheWAS figures (now Figures 4 & 5) and corrected the asthma classification.

Thank you for raising the discussion points about the interpretation of GRS associations with height and pack-years. We excluded lung function signals which were driven primarily by smoking behaviour (Methods and Supplementary Table 7), employing a Bonferroni threshold for 1020 SNPs tested across

any smoking trait. Whilst rigorous, this would not rule out very weak associations with smoking behaviour which, aggregated across the 372 SNPs comprising the FVC GRS, may explain the relatively weak association seen between the FVC GRS and pack-years. No association was seen between the GRS for the obstructive lung function traits, FEV₁ and FEV₁/FVC, and pack-years. We do not believe the observation of GRS associations with height is a sign that the phenotypes were not properly normalised or adjusted for. In the GWAS we adjusted for height and in sensitivity analyses of top SNPs, additional adjustments for height² and height³ provided similar findings (Results, end of 3rd paragraph p7). The relationship between lung function associations and height associations is not straightforward – we find individual lung function associated variants with opposite effects on height and we found that height relationships differed between different lung function traits (Discussion, penultimate paragraph p11). Our pathway pathway-partitioned GRS associations indicate that the relationship between genetic variants, height and lung function traits depends on the pathways through which the variants act and we include a statement in the discussion to that effect (end of penultimate paragraph p11).

Minor: maybe nobody reads supplementary materials anymore, but reading the ones for this paper requires major effort. Are all the figures necessary, or is the supplement a dumping ground for all analyses that were done (even if they yielded nothing interesting)?

We strongly advocate the reporting of all our methods and findings for transparency, to aid reproducibility, and as a resource to the scientific community. We recognise that different readers will wish to focus on different content within the Supplement and we have updated and carefully checked the indexing to aid such focused use, whilst provide concise summaries in the main paper.

Decision Letter, first revision:

8th September 2022

Dear Nick,

Your revised Article "Multi-ancestry genome-wide association study improves resolution of genes, pathways and pleiotropy for lung function and chronic obstructive pulmonary disease" has been seen by the original referees. You will see from their comments below that, while Reviewer #1 is satisfied with the revision and has no remaining requests, Reviewer #2 has a few ongoing concerns. We remain interested in the possibility of publishing your study in Nature Genetics, but we would like to consider your response to these ongoing concerns in the form of further revision before we make a final decision on publication.

As before, to guide the scope of the revisions, the editors discuss the referee reports in detail within the team, including with the chief editor, with a view to identifying key priorities that should be addressed in revision, and sometimes overruling referee requests that are deemed beyond the scope of the current study. In this case, we think Reviewer #2 makes a valid point that effect sizes will be overestimated, on average, in the discovery samples and that analyses based on these effect size estimates will be systematically biased. Accordingly, we ask that you further revise the presentation

and interpretation of these analyses taking this point into account. We again hope you will find this prioritized set of referee points to be useful when revising your study. Please do not hesitate to get in touch if you would like to discuss these issues further.

We therefore invite you to revise your manuscript again taking into account all reviewer and editor comments. Please highlight all changes in the manuscript text file. At this stage, we will need you to upload a copy of the manuscript in MS Word .docx or similar editable format.

*2) If you have not done so already please begin to revise your manuscript so that it conforms to our Article format instructions, available [here](http://www.nature.com/ng/authors/article_types/index.html). Refer also to any guidelines provided in this letter.

[redacted]

We again hope to receive your revised manuscript within 4-8 weeks. If you cannot send it within this time, please let us know.

Nature Genetics is committed to improving transparency in authorship. As part of our efforts in this direction, we are now requesting that all authors identified as 'corresponding author' on published

papers create and link their Open Researcher and Contributor Identifier (ORCID) with their account on the Manuscript Tracking System (MTS), prior to acceptance. ORCID helps the scientific community achieve unambiguous attribution of all scholarly contributions. You can create and link your ORCID from the home page of the MTS by clicking on 'Modify my Springer Nature account'. For more information please visit www.springernature.com/orcid.

Sincerely,
Kyle

Kyle Vogan, PhD
Senior Editor
Nature Genetics
<https://orcid.org/0000-0001-9565-9665>

Referee expertise:

Referee #1: Genetics, pulmonary diseases

Referee #2: Genetics, pulmonary diseases, statistical methods

Reviewers' Comments:

Reviewer #1:
Remarks to the Author:

In response to the reviewers' comments, the authors have substantially revised the manuscript, including the addition of several descriptions and statistical analyses. The figures have also been revised to make them very easy to understand. These changes have substantially improved the overall content.

Reviewer #2:
Remarks to the Author:

There are some specific comments from my first review that were not addressed properly in the revised version (they are repeated below). Again, estimated effect size based on a discovery GWAS are biased, with a bias that depends on allele frequency. The statements in the paper are incorrect and the small disclaimer in Discussion is not sufficient. For example, in "Effects in children", a correlation between the biased estimates of effect size in adults with the unbiased estimates in children is clearly biased. Why do the authors insist on presenting results that are clearly incorrect?

Comments that were not properly addressed.

There is a statement in the second paragraph of results that I am not convinced is true: "These 1020 signals show a pattern of increasing effect size as allele frequency decreases, in keeping with other complex traits". The sentence seems to imply that rare alleles tend to have higher effects (implying some sort of evolutionary selection on a trait that is related to lung function). I believe that this is just an outcome of "winner's curse": for the rare SNPs to make the significance cutoff, their sample effect size is a huge overestimation of the true effect size. Moreover, effect size are clearly different in different ancestry groups and what the authors call "effect size" is a weighted average of ancestry-specific effect sizes. This sentence needs to be rephrased or an accurate analysis of effect size needs to be performed.

A similar misinterpretation is in the "Effects in children" analysis. The authors compare effect sizes in children (calculated, it seems, on data from studies that were not used for GWAS) to effect sizes in adults (calculated from studies used in GWAS, and hence overestimated). The comparison is flawed because the adult data used in this comparison was part of the discovery analysis (if my understanding of which data was used is correct). You can see this clearly in Supplementary Figure 9 where the adult effect sizes tend to be larger than those in children (see above paragraph). Moreover, the correlation showed is also wrong (for the same reason) - in fact it might be underestimated. Note that this issue affects other analyses, for example what is shown in Supplementary Figure 10 (the issue there is that correlation is the improper statistic).

Author Rebuttal, first revision:

Response to Reviewers

Reviewer #1:

Remarks to the Author:

In response to the reviewers' comments, the authors have substantially revised the manuscript, including the addition of several descriptions and statistical analyses. The figures have also been revised to make them very easy to understand. These changes have substantially improved the overall content.

Reviewer #2:

Remarks to the Author:

There are some specific comments from my first review that were not addressed properly in the revised version (they are repeated below). Again, estimated effect size based on a discovery GWAS are biased, with a bias that depends on allele frequency. The statements in the paper are incorrect and the small disclaimer in Discussion is not sufficient. For example, in "Effects in children", a correlation between the biased estimates of effect size in adults with the unbiased estimates in children is clearly biased. Why do the authors insist on presenting results that are clearly incorrect?

We apologise that we misinterpreted these suggestions during the first revision and we thank the reviewer for these clarifications.

Comments that were not properly addressed.

There is a statement in the second paragraph of results that I am not convinced is true: "These 1020 signals show a pattern of increasing effect size as allele frequency decreases, in keeping with other complex traits". The sentence seems to imply that rare alleles tend to have higher effects (implying some sort of evolutionary selection on a trait that is related to lung function). I believe that this is just an outcome of "winner's curse": for the rare SNPs to make the significance cutoff, their sample effect size is a huge overestimation of the true effect size. Moreover, effect size are clearly different in different ancestry groups and what the authors call "effect size" is a weighted average of ancestry-specific effect sizes. This sentence needs to be rephrased or an accurate analysis of effect size needs to be performed.

We have removed this statement from the manuscript as it is not a key focus of the paper.

A similar misinterpretation is in the "Effects in children" analysis. The authors compare effect sizes in children (calculated, it seems, on data from studies that were not used for GWAS) to effect sizes in adults (calculated from studies used in GWAS, and hence overestimated). The comparison is flawed because the adult data used in this comparison was part of the discovery analysis (if my understanding of which data was used is correct). You can see this clearly in Supplementary Figure 9 where the adult effect sizes tend to be larger than those in children (see above paragraph). Moreover, the correlation showed is also wrong (for the same reason) - in fact it might be underestimated. Note that this issue affects other analyses, for example what is shown in Supplementary Figure 10 (the issue there is that correlation is the improper statistic).

At the reviewer's suggestion, we have undertaken a new analysis: (i) redefining sentinel SNPs from a GWAS in UK Biobank European ancestry individuals and (ii) then using effect size estimates for the sentinel SNPs from a meta-analysis of 34 studies of European ancestry adults (that is, independent of the

UK Biobank population used to identify these sentinels and therefore not subject to winner's curse bias). These unbiased adult effect size estimates were used to compare with effects in European-ancestry children (2 non-discovery cohorts), and this actually showed higher correlation coefficient estimates for 3 of the 4 lung function traits compared to our previous analyses (Supplementary Figure 7). We used the same unbiased adult effect size estimates to compare effect sizes between lung function and height (Supplementary Figure 9). We have updated the relevant sections of the Methods, Results, Discussion and Supplementary Figures 7 and 9 accordingly.

Decision Letter, second revision:

21st October 2022

Dear Nick,

Your revised manuscript "Multi-ancestry genome-wide association study improves resolution of genes, pathways and pleiotropy for lung function and chronic obstructive pulmonary disease" (NG-A60012R1) has been seen by Reviewer #2. As you will see from the comments below, Reviewer #2 is satisfied with the revision and has no remaining requests, and therefore we will be happy in principle to publish your study in Nature Genetics as an Article pending final revisions to comply with our editorial and formatting guidelines.

We are now performing detailed checks on your paper, and we will send you a checklist detailing our editorial and formatting requirements soon. Please do not upload the final materials or make any revisions until you receive this additional information from us.

Thank you again for your interest in Nature Genetics. Please do not hesitate to contact me if you have any questions.

Sincerely,
Kyle

Kyle Vogan, PhD
Senior Editor
Nature Genetics
<https://orcid.org/0000-0001-9565-9665>

Referee expertise:

Referee #2: Genetics, pulmonary diseases, statistical methods

Reviewer #2 (Remarks to the Author):

The changes the authors made are acceptable. I am satisfied now.

Final Decision Letter:

25th January 2023

Dear Nick,

I am delighted to say that your manuscript "Multi-ancestry genome-wide association analyses improve resolution of genes and pathways influencing lung function and chronic obstructive pulmonary disease risk" has been accepted for publication in an upcoming issue of Nature Genetics.

Your paper will be published online after we receive your corrections and will appear in print in the next available issue. You can find out your date of online publication by contacting the Nature Press Office (press@nature.com) after sending your e-proof corrections. Now is the time to inform your Public Relations or Press Office about your paper, as they might be interested in promoting its publication. This will allow them time to prepare an accurate and satisfactory press release. Include your manuscript tracking number (NG-A60012R2) and the name of the journal, which they will need when they contact our Press Office.

Before your paper is published online, we will be distributing a press release to news organizations worldwide, which may very well include details of your work. We are happy for your institution or funding agency to prepare its own press release, but it must mention the embargo date and Nature Genetics. Our Press Office may contact you closer to the time of publication, but if you or your Press Office have any enquiries in the meantime, please contact press@nature.com.

Acceptance is conditional on the data in the manuscript not being published elsewhere, or announced in the print or electronic media, until the embargo/publication date. These restrictions are not intended to deter you from presenting your data at academic meetings and conferences, but any

enquiries from the media about papers not yet scheduled for publication should be referred to us.

Please note that Nature Genetics is a Transformative Journal (TJ). Authors may publish their research with us through the traditional subscription access route or make their paper immediately open access through payment of an article-processing charge (APC). Authors will not be required to make a final decision about access to their article until it has been accepted. [Find out more about Transformative Journals](https://www.springernature.com/gp/open-research/transformative-journals)

Authors may need to take specific actions to achieve [compliance](https://www.springernature.com/gp/open-research/funding/policy-compliance-faqs) with funder and institutional open access mandates. If your research is supported by a funder that requires immediate open access (e.g. according to [Plan S principles](https://www.springernature.com/gp/open-research/plan-s-compliance)), then you should select the gold OA route, and we will direct you to the compliant route where possible. For authors selecting the subscription publication route, the journal's standard licensing terms will need to be accepted, including [self-archiving-and-license-to-publish](https://www.nature.com/nature-portfolio/editorial-policies/self-archiving-and-license-to-publish). Those licensing terms will supersede any other terms that the author or any third party may assert apply to any version of the manuscript.

Please note that Nature Portfolio offers an immediate open access option only for papers that were first submitted after 1 January 2021.

If you have not already done so, we invite you to upload the step-by-step protocols used in this

manuscript to the Protocols Exchange, part of our on-line web resource, natureprotocols.com. If you complete the upload by the time you receive your manuscript proofs, we can insert links in your article that lead directly to the protocol details. Your protocol will be made freely available upon publication of your paper. By participating in natureprotocols.com, you are enabling researchers to more readily reproduce or adapt the methodology you use. Natureprotocols.com is fully searchable, providing your protocols and paper with increased utility and visibility. Please submit your protocol to <https://protocolexchange.researchsquare.com/>. After entering your nature.com username and password you will need to enter your manuscript number (NG-A60012R2). Further information can be found at <https://www.nature.com/nature-portfolio/editorial-policies/reporting-standards#protocols>

Sincerely,
Kyle

Kyle Vogan, PhD
Senior Editor
Nature Genetics
<https://orcid.org/0000-0001-9565-9665>